# Dichotomous outcomes of TNFR1 and TNFR2 signaling in NK cell-mediated immune responses during inflammation

Timothy R. McCulloch [1,2,10] ✉, Gustavo R. Rossi[1,10], Louisa Alim [1,10], Pui Yeng Lam [1], Joshua K. M. Wong[1], Elaina Coleborn [1], Snehlata Kumari [1], Colm Keane[1,3], Andrew J. Kueh[4,5], Marco J. Herold [4,5,6], Christoph Wilhelm [2], Percy A. Knolle [7], Lawrence Kane [8], Timothy J. Wells [1,9] & Fernando Souza-Fonseca-Guimaraes [1] ✉

Natural killer (NK) cell function is regulated by a balance of activating and inhibitory signals. Tumor necrosis factor (TNF) is an inflammatory cytokine ubiquitous across homeostasis and disease, yet its role in regulation of NK cells remains unclear. Here, we find upregulation of the immune checkpoint protein, T cell immunoglobulin and mucin domain 3 (Tim3), is a biomarker of TNF signaling in NK cells during *Salmonella* Typhimurium infection. In mice with conditional deficiency of either TNF receptor 1 (TNFR1) or TNF receptor 2 (TNFR2) in NK cells, we find TNFR1 limits bacterial clearance whereas TNFR2 promotes it. Mechanistically, via single cell RNA sequencing we find that both TNFR1 and TNFR2 induce the upregulation of Tim3, while TNFR1 accelerates NK cell death but TNFR2 promotes NK cell accumulation and effector function. Our study thus highlights the complex interplay of TNF-based regulation of NK cells by the two TNF receptors during inflammation.

Natural killer (NK) cells are circulating innate lymphocytes that play important roles in host defense by promoting immunity against viral infections[1] and cancer[2]. NK cells also have the potential to contribute to anti-bacterial immunity through their ability to directly kill bacteria or infected cells, as well as produce important cytokines such as interferon-gamma (IFN-γ), tumor necrosis factor (TNF), and granulocyte-macrophage colony-stimulating factor (GM-CSF), among others[3–5]. For example, NK cells have been shown to be an important contributor to protective IFN-γ production during *Salmonella* Typhimurium infection[4]. Further, invasive bacterial infection, bacterial sepsis, and bacterial pneumonia are more frequent in patients with NK cell

deficiency, suggesting NK cells play a protective role[6]. However, recent studies have suggested that certain inhibitory receptors and/or cytokines may impair the effector function of NK cells, promoting the progression and persistence of disease[7].

One such cytokine that has the potential to impact NK cell function is TNF, a highly pleiotropic cytokine that is expressed across homeostasis and disease[8]. How TNF acts on NK cells to govern their function remains poorly defined[9]. TNF signals through two receptors, TNF receptor 1 (TNFR1) and TNF receptor 2 (TNFR2). TNFR1 expression is relatively ubiquitous across cell types, whereas TNFR2 expression is mainly constrained to immune lineages including NK cells, as well as

[1]Frazer Institute, The University of Queensland, Woolloongabba, Australia. [2]Institute of Clinical Chemistry and Clinical Pharmacology, University Hospital Bonn, University of Bonn, Bonn, Germany. [3]Princess Alexandra Hospital, Woolloongabba, Australia. [4]The Walter and Eliza Hall Institute of Medical Research, Parkville, Australia. [5]Department of Medical Biology, University of Melbourne, Parkville, Australia. [6]Olivia Newton-John Cancer Research Institute, Heidelberg, Australia. [7]Institute of Molecular Immunology, School of Medicine and Health, Technical University of Munich, Munich, Germany. [8]Department of Immunology, University of Pittsburgh School of Medicine, Pittsburgh, PA, USA. [9]Australian Infectious Diseases Research Centre, University of Queensland, Brisbane, Australia. [10]These authors contributed equally: Timothy R. McCulloch, Gustavo R. Rossi, Louisa Alim. ✉e-mail: timothym@uni-bonn.de; f.guimaraes@uq.edu.au

oligodendrocytes and endothelial cells[10]. In T cells, TNF has been associated with dysfunction during chronic viral infection and cancer through upregulation of programmed cell death protein 1 (PD-1) and T cell immunoglobulin and mucin domain-containing protein 3 (Tim3), as well as impairment of cytokine production[11,12]. In a model of colitis, TNFR2 signaling can induce regulatory T cell (Treg) differentiation and enhance their suppressive activity[13]. Interestingly, in other colitis models, TNF can also promote cytokine production in conventional CD4⁺ T cells through TNFR2, exacerbating autoimmunity[14]. Whereas TNFR1 is yet to be investigated on NK cells, TNFR2 expression on NK cells has been shown to be an important factor for IFN-γ production by NK cells in response to dendritic cell (DC) crosstalk[15,16]. Thus, the role of TNF on lymphocyte responses is complex and likely context, cell type, and pathology-dependent. More research is required to fully appreciate how TNF impacts NK cell function across different contexts, including bacterial infection.

In the present study, we use *S.* Typhimurium as a model to explore the role of TNF on NK cells. We identify TNF signaling as a major driver of Tim3 expression in NK cells during infection. We further find that TNF signaling through either TNFR1 or TNFR2 plays both overlapping as well as unique roles in regulating NK cell function, where TNFR1 impairs but TNFR2 supports NK cell accumulation and function. These results underpin the complexity of TNF signaling on NK cell survival, accumulation, and effector function, and highlight potential new avenues to target and improve efficacy of NK cell-based immunotherapies.

## Results

### The checkpoint inhibitor Tim3 is upregulated on effector NK cells during bacterial infection

We have previously shown that NK cells are significantly impaired during bacterial infection, including differentiation into ILC1-like cells and organ-specific depletion[17]. Here, by infecting C57BL/6 mice with attenuated *S.* Typhimurium SL3261 (causing a chronic sublethal infection[18]) and using flow cytometry to phenotype the NK cells from spleen and liver (gating strategy shown in Supplementary Fig. 1a), we find that bacterial infection also induces robust maturation (Supplementary Fig. 1b–d) and proliferation (Supplementary Fig. 1e–g) in both organs. However, the precise mechanisms underlying the regulation of NK cells during bacterial infection are currently not well understood. To further investigate the regulation of NK cells during bacterial infection, we performed single cell-RNA-sequencing (scRNA-seq) on NK cells sorted from the spleens of *S.* Typhimurium infected mice (sorting strategy and purity shown in Supplementary Fig. 2a, b). The scRNA-seq analysis identified six distinct clusters of NK cells according to effector functions, maturation subsets, and proliferative stages (Fig. 1a). Using signature gene expression, we identified mature NK cells (cluster 0) expressing *Itgam* and *Klrg1*, activated/effector NK cells (cluster 3) expressing cytotoxic genes such as *Gzma*, *Gzmb*, *Gzmc*, and *Prf1*, as well as *Ifng*, less mature NK cells (cluster 5) expressing *Cd7*, *Tcf7*, and *Tox*, and proliferating NK cells (clusters 1, 2, and 4) expressing *Mki67* and various cell cycle genes (Fig. 1b). Cell cycle scoring predicted that the different proliferative clusters were in different stages of the cell cycle, where cluster 2 was mainly in S phase, and clusters 1 and 4 predominantly G2/M phase (Supplementary Fig. 2c). Interestingly, the activated/effector NK cells (cluster 3) which expressed high levels of *Ifng*, *Gzmc*, and *Il12rb1* also showed increased expression of *Havcr2*, which encodes for the inhibitory checkpoint molecule Tim3 (Fig. 1c). Expression of *Havcr2* appeared to correlate well with the expression of effector molecules including *Ifng*, as well as *Thy1*, and *Zbtb32* (Fig. 1d). *Thy1* encodes Thy1, which has been shown to be a marker of the effector NK cell population which contributes to immunity to *S.* Typhimurium infection[4], and *Zbtb32* encodes for the transcription factor Zbtb32, which is essential for the proliferative burst of NK cells in response to infection[19]. We validated the upregulation of Tim3 by flow cytometry and observed that Tim3 was elevated

on NK cells in both the spleen (Fig. 1e, f) and liver (Supplementary Fig. 2d, e) of *S.* Typhimurium infected mice compared to naive controls. Moreover, the expression of Tim3 was highest on day four post-infection and gradually decreases as the infection resolved by day 28 (Fig. 1g). Expression of another member of the Tim family, Tim4 (encoded by *Timd4*), was not found to be upregulated in NK cells at either the gene or protein level (Supplementary Fig. 3a–c), suggesting this upregulation was specific for Tim3. Taken together, these results suggest that the upregulation of Tim3 in activated/effector NK cells during early *S.* Typhimurium infection may be a mechanism that regulates NK cell effector function and impedes anti-bacterial immunity.

### Physiological levels of Tim3 do not impact NK cell function during *S.* Typhimurium infection

To investigate the impact of Tim3 expression on NK cells, we generated transgenic mouse lines with conditional deletion (*Ncr1^{cre}Tim3^{fl/fl}*) or overexpression (*Ncr1^{cre}FSF-Tim3*) of Tim3 specifically within NK cells. The development of NK cells and ILC1 appeared mostly normal in each strain (Supplementary Fig. 4a–e), except for a slight decrease in the percentage of NK cells in the intermediate M1 maturation stage across all organs examined (spleen NK cells shown as representative in Supplementary Fig. 4f). However, functional differences were observed between the transgenic mice. Overexpression of Tim3 led to significant reductions in both NK cell IFN-γ expression and degranulation (measured by CD107a staining) compared to wild-type controls when stimulated with plate-bound anti-NK1.1 antibodies (Fig. 2a, b). This reduction in IFN-γ was also observed in IL-12/IL-18 stimulated cells (Fig. 2a). In adoptive transfer experiments, by transferring a 50:50 mix of Tim3-deficient and Tim3-overexpressing NK cells (Fig. 2c) we found that deletion of Tim3 lead to significant improvements in NK cell accumulation in the spleen and liver compared to Tim3 overexpressing cells, both in naïve mice and in the context of *S.* Typhimurium infection (Fig. 2d and Supplementary Fig. 5a, b). To assess how Tim3 expression modulated NK cell function in vivo, *Ncr1^{cre}Tim3^{fl/fl}*, *Ncr1^{cre}FSF-Tim3*, and wild-type *Ncr1^{cre}* controls were infected with *S.* Typhimurium and spleens and livers were taken at day four post-infection to observe immune parameters and bacterial burdens. Tim3 deletion and overexpression were confirmed in this model by flow cytometry staining (Supplementary Fig. 5c, d). While we found that *Ncr1^{cre}FSF-Tim3* had reduced NK cell numbers (Fig. 2e, f), NK cell activation (by CD69 staining, Fig. 2g, h), and increased bacterial burdens in the spleen (Fig. 2i, j), we did not observe the expected increases in NK cell numbers (Fig. 2e, f), changes in bacterial burdens (Fig. 2i, j), or differences in IFN-γ (Fig. 2k) in *Ncr1^{cre}Tim3^{fl/fl}* mice compared to controls. In fact, NK cells were surprisingly reduced in the livers of *Ncr1^{cre}Tim3^{fl/fl}* mice (Fig. 2f). Further, blockade of Tim3 with monoclonal antibodies could not reduce bacterial burdens (Supplementary Fig. 5e, f). Taken together, these data suggest that while overexpression of Tim3 can impact NK cell accumulation and function, Tim3 at physiological levels does not significantly impact NK cell-mediated control of *S.* Typhimurium infection.

### TNF induces Tim3 expression and proliferation but inhibits NK cell survival

Previous reports have shown that Tim3 can be upregulated on human NK cells[20] and murine T cells[12] in response to TNF. Similarly, type I interferons (IFNs) and transforming growth factor (TGF)-β have also been shown to drive Tim3 upregulation in T cells[21] and macrophages[22], respectively. Thus, we predicted that Tim3 expression could be a biomarker for other factors impacting NK cell function in our model. To investigate potential cytokine drivers of Tim3 expression in murine NK cells, we cultured murine NK cells with recombinant IL-15 and various concentrations of TNF, IFN-α, and TGF-β, alone or in combination. We found that increasing levels of TNF increased the expression of Tim3 in a stepwise manner, similar to what has been observed

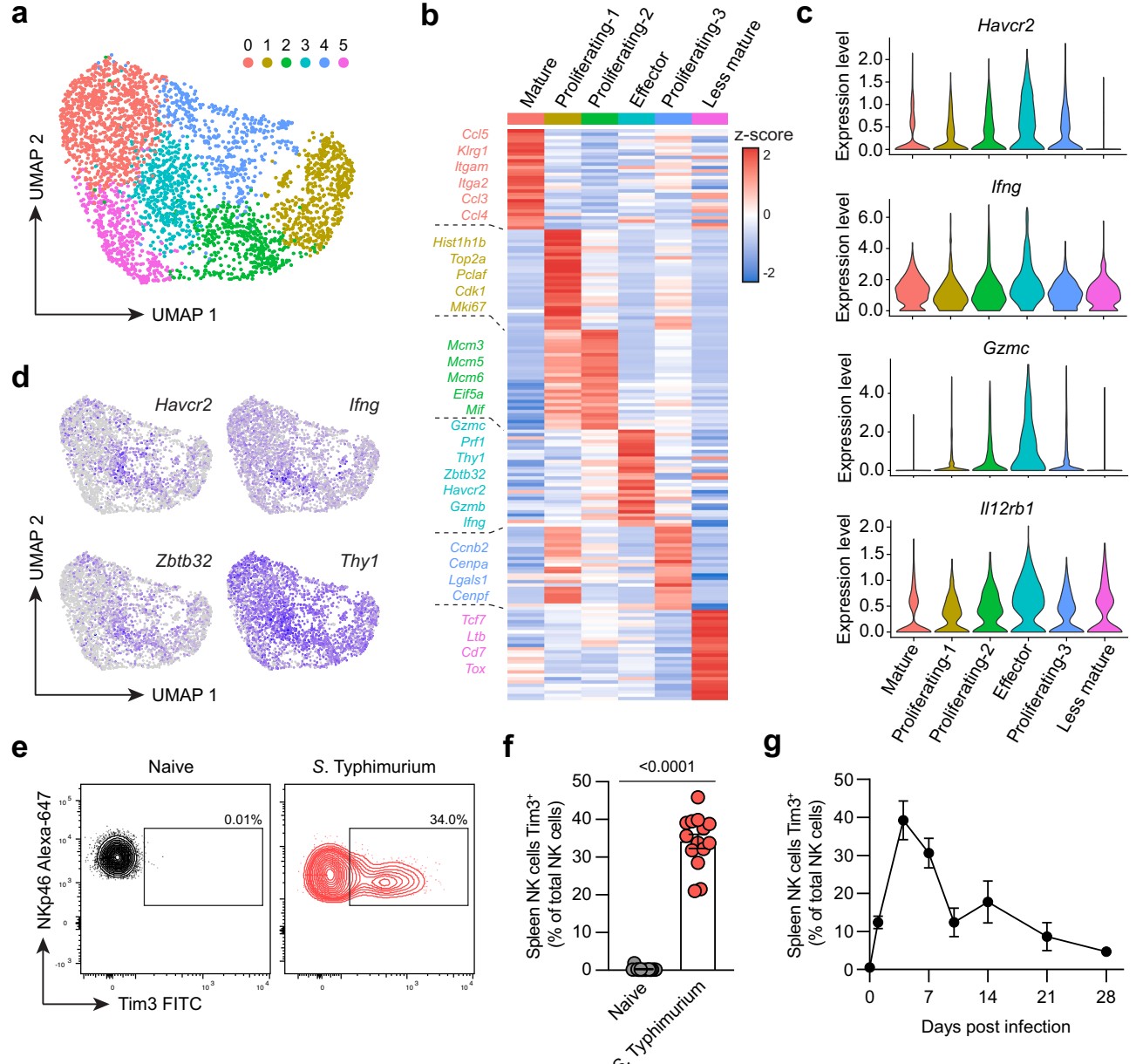

**Fig. 1 | scRNA-sequencing identifies Tim3 upregulation in effector NK cells during *S.* Typhimurium infection.** Wild-type C57BL/6 mice were infected with *S.* Typhimurium, and NK cells were analyzed by scRNA-seq or flow cytometry on day four post-infection. **a** UMAP plot of NK cells colored by Seurat clusters, where each dot represents a single cell. **b** Heatmap showing the top 20 signature genes identifying Seurat clusters. **c** Violin plots showing expression of indicated genes across Seurat clusters. **d** UMAP plots showing normalized expression of *Ifng, Zbtb32, Havcr2,* and *Thy1*. **e** Representative flow cytometry plots showing expression of Tim3 in splenic NK cells from naïve mice or day four post *S.* Typhimurium infection.

**f** Percentage of NK cells expressing Tim3 in the spleen of naïve or *S.* Typhimurium infected mice. **g** Percentage of NK cells expressing Tim3 in the spleen of mice at different time points post *S.* Typhimurium infection. Data in (**a–d**) from a single experiment (*n* = 4 biological replicates) and (**e, f**) pooled from two independent experiments (*n* = 15 naïve mice and 14 *S.* Typhimurium infected). Data in (**g**) is representative of two independent experiments (*n* = 4-5 biological replicates). Error bars indicate mean ± SEM. Groups were compared using a two-tailed Mann-Whitney *U* test, where *P* < 0.05 was considered statistically significant. Source data are provided as a Source Data file.

in human NK cells (Fig. 3a). In contrast, IFN-α had no effect on Tim3 expression. Adding TGF-β reduced Tim3 expression, and when added in combination with TNF, completely prevented TNF-induced Tim3 upregulation (Fig. 3a). We also observed that NK cell-related activating cytokines IL-12 and IL-18 were able to upregulate Tim3 (Supplementary Fig. 6a). TNF acting via its receptors can activate multiple distinct signaling pathways, including canonical nuclear factor kappa-light-chain-enhancer of activated B cells (NF-κB), non-canonical NF-κB, and mitogen-activated protein kinases (MAPKs). To assess which of these pathways were required for Tim3 upregulation, NK cells were cultured with TNF in the presence of various TNF pathway inhibitors. Targeting

canonical NF-κB with a nuclear factor kappa-B kinase beta (IKK-β) inhibitor (BI605906[23]) was able to prevent TNF-induced Tim3 upregulation in a dose-dependent manner (Supplementary Fig. 6b). Conversely, targeting MAPKs c-Jun N-terminal kinase (JNK, inhibited by SP600125[24]) and p38 (inhibited by Losmapimod[25]) showed no effect, nor did targeting non-canonical NF-κB with a NF-κB-inducing kinase (NIK) inhibitor (Amgen16[26], Supplementary Fig. 6c−e). mTOR inhibitor Rapamycin was also able to prevent Tim3 upregulation (Supplementary Fig. 6f), likely due to the critical importance of mTOR in IL-15-mediated NK cell activation[27]. Thus, these data show that TNF induces Tim3 upregulation via the canonical NF-κB pathway. We also used CTV

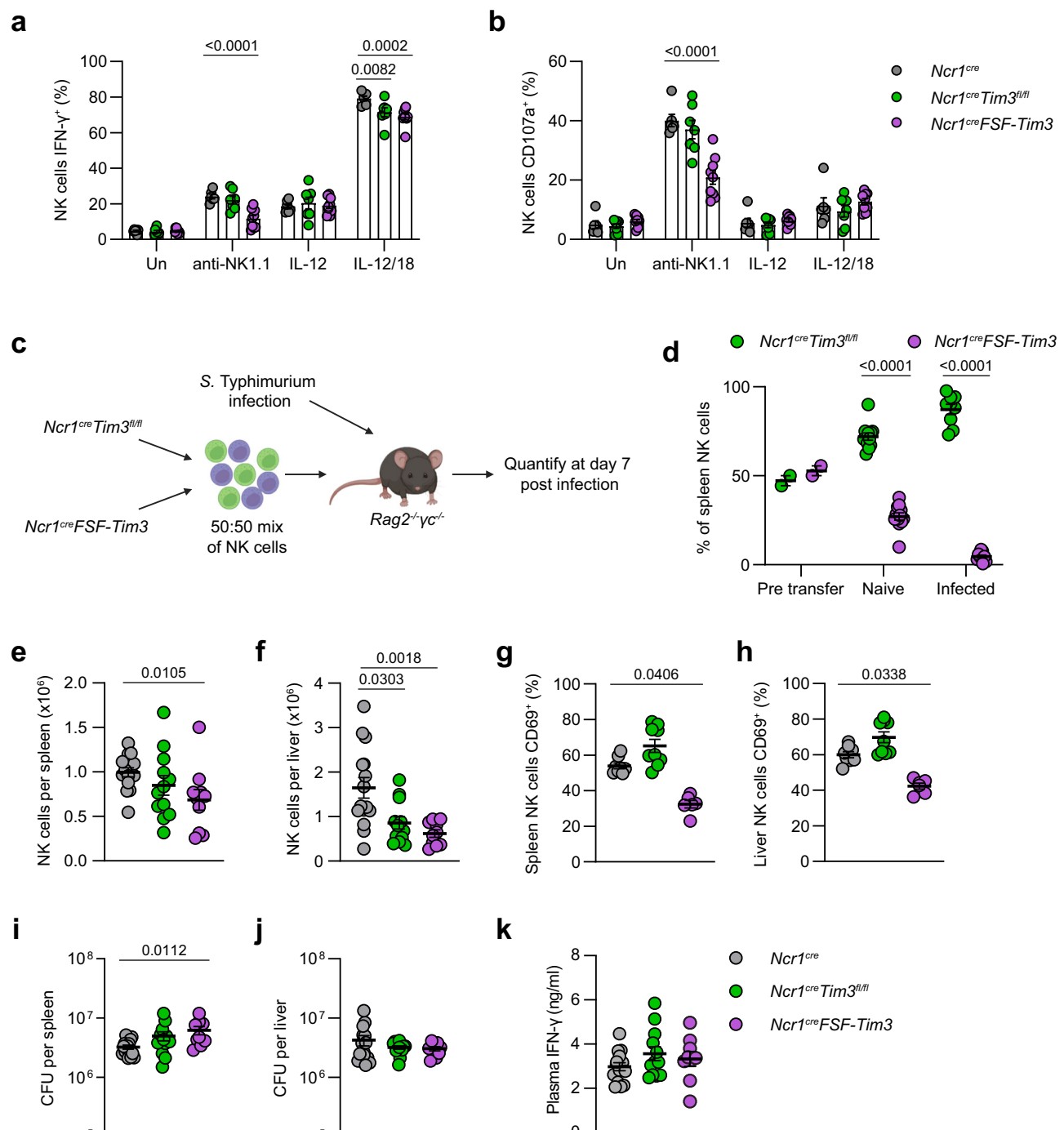

**Fig. 2 | Overexpression of Tim3 restricts NK cell function and accumulation.** Splenocytes from *Ncr1cre* (wild-type), *Ncr1creTim3fl/fl* (Tim3-null), or *Ncr1creFSF-Tim3* (Tim3-overexpression) mice were cultured for four hours with 50 ng/ml IL-15 and the indicated stimulations; plate bound anti-NK1.1, recombinant IL-12 (25 ng/ml), and/or recombinant IL-18 (5 ng/ml). **a**, **b** Percentage of NK cells expressing intracellular IFN-γ (**a**) or cell surface CD107a (**b**). NK cells isolated from Tim3-null or Tim3-overexpressing mice were adoptively transferred into *Rag2⁻/⁻γc⁻/⁻* mice and recipient mice infected with *S.* Typhimurium or left naïve. **c** Schematic showing the experimental procedure. Created in BioRender. McCulloch, T. (2023) BioRender.com/g40v791. **d** Relative proportions of NK cells pre-transfer and in spleens of adoptively transferred mice with or without infection. *Ncr1cre*, *Ncr1creTim3fl/fl*, and *Ncr1creFSF-Tim3* mice were infected with *S.* Typhimurium, and spleens and livers were analyzed at day 4 post-infection. **e**, **f** Total numbers of NK cells isolated from the spleen (**e**) or liver (**f**). **g**, **h** Expression of CD69 on spleen (**g**) or liver (**h**) NK cells. **i**, **j** Bacterial burden per spleen (**i**) or liver (**j**), determined by CFU counts. **k** Titers of IFN-γ from plasma of infected mice. Data pooled from two independent experiments (**a**, **b**: $n = 6$ *Ncr1cre*, 7 *Ncr1creTim3fl/fl*, and 9 *Ncr1creFSF-Tim3*; **c**, **d**: $n = 12$ naïve mice and 9 *S.* Typhimurium infected; **e**–**k**: $n = 15$ *Ncr1cre*, 12 *Ncr1creTim3fl/fl*, and 9 *Ncr1creFSF-Tim3*). Error bars indicate mean ± SEM. Groups were compared using the Kruskal-Wallis test with Dunn's multiple comparisons test for (**a**, **b**) and (**e**–**k**), or the two-tailed Mann-Whitney U test for (**d**), where $P < 0.05$ was considered statistically significant. Abbreviations: IFN, interferon; CFU, colony forming units. Source data are provided as a Source Data file.

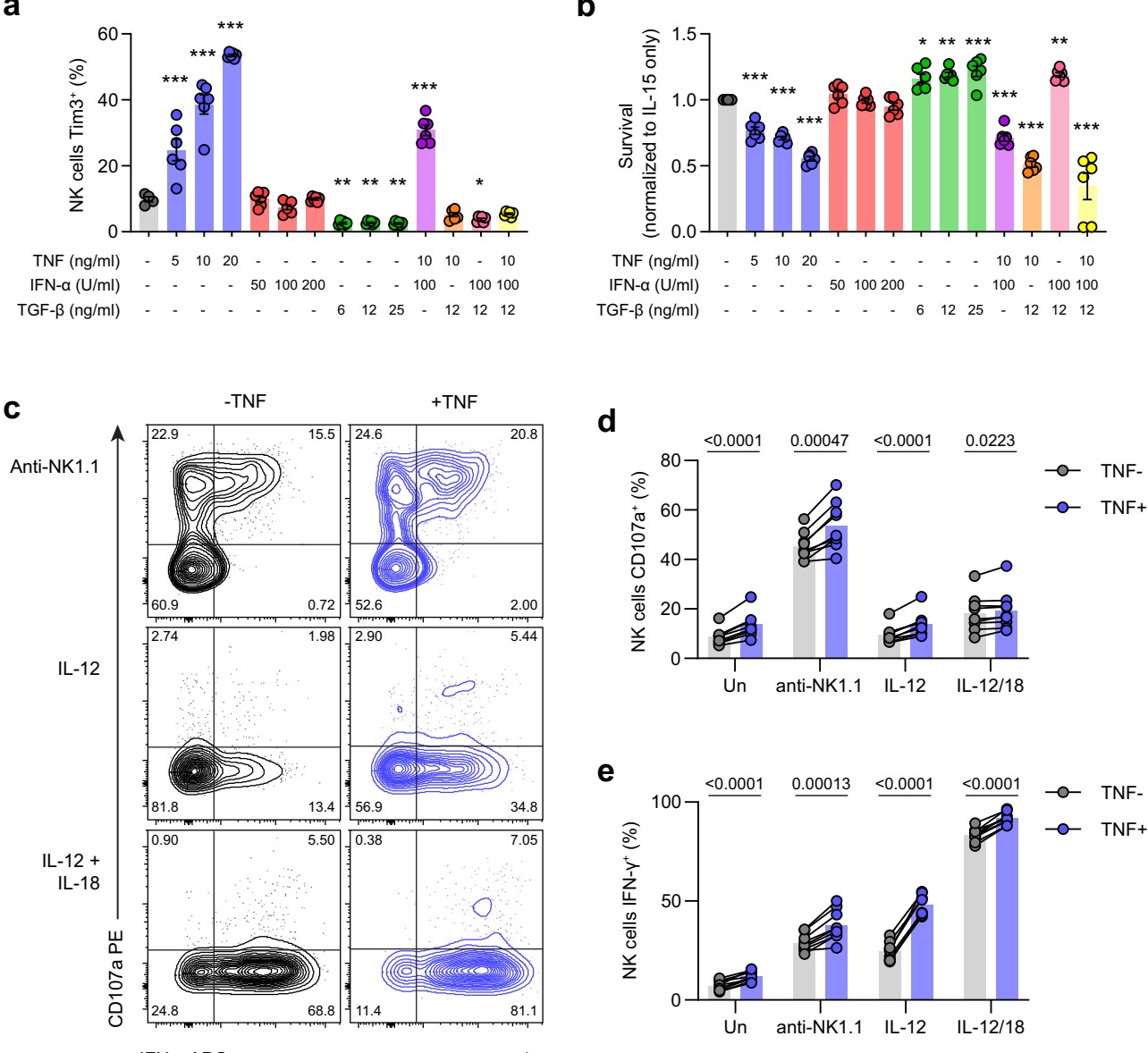

**Fig. 3 | TNF upregulates Tim3 and impacts NK cell function and survival in vitro.** NK cells from wild-type C57BL/6 mice were isolated and cultured for three days in the presence of the 10 ng/ml IL-15 plus the indicated cytokines. **a** Expression of Tim3 on NK cells. **b** Overall survival of NK cells, normalized to IL-15 only. Total splenocytes from C57BL/6 mice were cultured for four hours in the presence of the indicated stimulations with or without 10 ng/ml recombinant TNF. **c** Representative flow cytometry plots showing expression of CD107a and IFN-γ in NK cells. **d**, **e** Percentage of NK cells expressing IFN-γ (**d**) or CD107a (**e**). Data are from two

independent experiments (**a**, **b**: *n* = 6 biological replicates, **c**–**e**: *n* = 9 biological replicates). Each dot represents one animal, and error bars indicate mean ± SEM. Groups were compared using one-way ANOVA with Šídák's multiple comparisons test for (**a**, **b**) (comparisons to IL-15 only, *$P < 0.05$, **$P < 0.01$, ***$P < 0.001$) or paired *t* test for **c**, **d** where $P < 0.05$ was considered statistically significant. Abbreviations: TNF, tumor necrosis factor; IFN, interferon; TGF, tumor growth factor. Source data are provided as a Source Data file.

dye dilution to quantify NK cell proliferation and survival, as described previously[28]. TNF at either 5 or 10 ng/ml was able to promote the division of NK cells (Supplementary Fig. 6g, h), indicating enhanced proliferation. This effect was lost with the addition of TGF-β, which completely ablated proliferation (Supplementary Fig. 6h). Increasing levels of TNF also led to significant reduction in NK cell survival compared to IL-15-only culture, whereas IFN-α once again had no effect (Fig. 3b). Interestingly, the addition of TGF-β could not prevent TNF-induced cell death, suggesting an independent pathway compared to Tim3 upregulation and proliferation.

To investigate if TNF could also promote NK cell effector function, we stimulated splenocytes with plate-bound anti-NK1.1 antibody, IL-12, or IL-12 and IL-18 for four hours in the presence or absence of

recombinant murine TNF and measured degranulation and IFN-γ production (Fig. 3c). Addition of TNF significantly enhanced CD107a staining and IFN-γ expression in NK cells across all culture conditions, particularly in IL-12 stimulated cultures (Fig. 3d, e). Together, these data suggest TNF can induce dysfunction in NK cells as well as promote their effector function.

## TNF signaling alters NK cell phenotype during bacterial infection
Reanalysis of the scRNA-seq from Fig. 1 found that NK cells express both receptors for TNF, TNFR1 (encoded by *Tnfrsf1a*) and TNFR2 (encoded by *Tnfrsf1b*), although there did not appear to be differences in expression patterns across the clusters (Supplementary Fig. 7a, b).

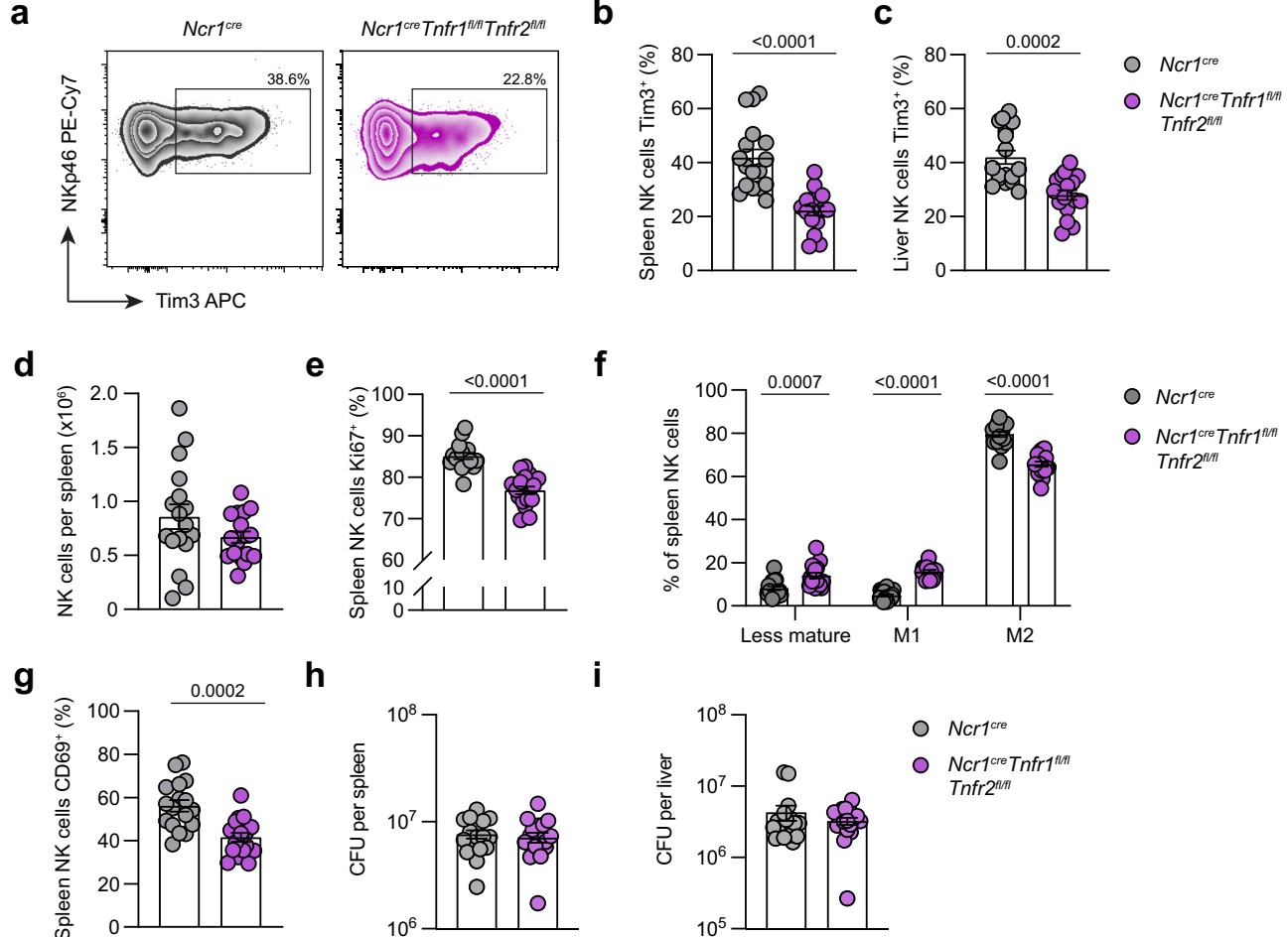

**Fig. 4 | Suppression of TNF signaling in NK cells impacts their phenotype during infection.** Transgenic *Ncr1*[cre] and *Ncr1*[cre]*Tnfr1*[fl/fl]*Tnfr2*[fl/fl] mice were infected with *S.* Typhimurium, and spleens and livers analyzed on day 4 post-infection. **a** Representative flow cytometry plots showing expression of Tim3 on splenic NK cells. **b**, **c** Percentage of spleen (**b**) or liver (**c**) NK cells expressing Tim3. **d** Total count of NK cells in the spleen of infected mice. **e** Percentage of NK cells expressing Ki67. **f** Percentage of splenic NK cells in each maturation stage based on CD11b and KLRG1

expression (Less mature = CD11b⁻KLRG1⁻, M1 = CD11b⁺KLRG1⁻, M2 = CD11b⁺KLRG1⁺). **g** Percentage of NK cells expressing CD69. **h**, **i** Bacterial burden per spleen (**h**) or liver (**i**), determined by CFU counts. Data are from two independent experiments (*n* = 17 biological replicates). Each dot represents one animal, and error bars indicate mean ± SEM. Groups were compared using a two-tailed Mann-Whitney *U* test, where *P* < 0.05 was considered statistically significant. Abbreviations: CFU, colony forming units. Source data are provided as a Source Data file.

This was confirmed in naïve mice by flow cytometry, where we found no difference in the expression pattern of either receptor across the different maturation states (Supplementary Fig. 7c, d). To assess the dynamics of TNFR expression on NK cells during *S.* Typhimurium infection, we examined the expression of TNFR1 and TNFR2 over the course of infection. While TNFR1 expression dropped at day 7-10 and recovered at day 14 (Supplementary Fig. 8a), TNFR2 expression increased dramatically during early infection before gradually returning to baseline (Supplementary Fig. 8b), showing inflammatory signals could modulate the expression of these receptors in NK cells. This data is consistent with a recent study showing LCMV infection and cytokines IL-12 and IL-18 could upregulate TNFR2 on NK cells[29]. Cytokine analysis in the serum of these animals showed that TNF peaked early during infection (Supplementary Fig. 8c), whereas IFN-γ peaked at day 7-10 before returning to baseline by day 21 (Supplementary Fig. 8d). In addition, we observed in an oral infection model that TNFR1 expression remained unchanged (Supplementary Fig. 8e, f) while TNFR2 expression was increased in NK cells during infection (Supplementary Fig. 8g, h). Tim3 was also upregulated on splenic NK cells following oral infection (Supplementary Fig. 8i, j), further corroborating our findings.

To further examine how TNF impacts NK cells in vivo, we generated a novel mouse model of TNF signaling deficiency, specifically in

NK cells (*Ncr1*[cre]*Tnfr1*[fl/fl]*Tnfr2*[fl/fl]). Following infection of these mice, deletion of TNF signaling significantly reduced the expression of Tim3 on NK cells, both in the spleen and liver (Fig. 4a–c), confirming that TNF is a driver of Tim3 upregulation during *S.* Typhimurium infection. Although there was no difference in NK cell numbers in the TNFR-floxed mice (Fig. 4d), we saw a reduction in expression of proliferation marker Ki67 (Fig. 4e). Deletion of the TNF receptors also led to a reduction in terminal maturation of the NK cells (Fig. 4f) as well as a reduction in activation, determined by CD69 expression (Fig. 4g). However, despite these phenotypic changes to NK cells, we found no differences in bacterial burdens in the conditional knockout compared to wild-type controls in either the spleen or liver (Fig. 4h, i). Thus, while TNF receptor signaling impacts NK phenotypes in vivo, their impact on NK cell effector function and anti-bacterial capacity is unclear.

## TNFR1 and TNFR2 contribute to NK cell maturation

We hypothesized the lack of anti-bacterial phenotype in the *Ncr1*[cre]*Tnfr1*[fl/fl]*Tnfr2*[fl/fl] mice could be due to TNFR1 and TNFR2 playing opposing roles on NK cell function. To investigate the contribution of each receptor independently, we generated novel transgenic mouse lines with conditional deletion of TNFR1 (*Ncr1*[cre]*Tnfr1*[fl/fl]) or TNFR2 (*Ncr1*[cre]*Tnfr2*[fl/fl]) specifically within NK cells. In each mouse line, NK cell

and ILC1 numbers were normal (Supplementary Fig. 9a–c), and TNFR1 or 2 expression was confirmed to be deleted in the appropriate models (Supplementary Fig. 9d, e). NK cell development appeared normal except for slight decreases in the terminal maturation in both transgenic models across all organs examined (spleen shown as representative in Supplementary Fig. 9f).

To investigate how these deletions impacted NK cell function during infection, $Ncr1^{cre}Tnfr1^{fl/fl}$, $Ncr1^{cre}Tnfr2^{fl/fl}$, or $Ncr1^{cre}$ controls were infected with S. Typhimurium and splenic NK cells isolated at day four post-infection for scRNA-seq. As with the WT scRNA-seq in Fig. 1, we were able to identify six separate NK cell clusters (Fig. 5a). These clusters conformed to similar identities to the wild-type scRNA-seq, where cluster 0 was mature, cluster 1 was less mature, cluster 4 was effector/activated, and clusters 2, 3, and 5 were proliferative (Fig. 5b and Supplementary Fig. 10a). As with the scRNA-seq from Fig. 1, the different proliferative clusters were predicted to be in different stages of the cell cycle, where cluster 3 was mostly assigned to S phase and clusters 2 and 5 predominantly assigned to G2/M phase (Supplementary Fig. 10b).

To examine compositional changes in NK cell phenotypes upon deletion of either TNFR1 or TNFR2, we compared the differential abundance of cellular neighborhoods using Milo[30] (Fig. 5c–f). We found that most neighborhoods within the less mature cluster (1) were significantly enriched in both $Ncr1^{cre}Tnfr1^{fl/fl}$ (Fig. 5c, d) and $Ncr1^{cre}Tnfr2^{fl/fl}$ (Fig. 5e, f) NK cells compared to WT, suggesting deleting either TNFR1 or TNFR2 limited NK cell maturation. We also saw an increased representation of neighborhoods within the proliferating clusters of WT NK cells compared to $Ncr1^{cre}Tnfr2^{fl/fl}$, suggesting that TNFR2 may contribute to NK cell proliferation (Fig. 5e, f). When comparing the relative frequency of cells within each cluster by genotype, we found that there was an increase in the less mature cluster and decrease in the combined proliferating clusters in both the TNFR1 and TNFR2 deficient NK cells (Supplementary Fig. 10c, d). We also observed a reduction in the expression of terminal maturation genes, such as Klrg1, as well as increases in the expression of genes associated with less mature NK cells, such as Cd7 (Supplementary Fig. 10e). We confirmed this phenotype by flow cytometry of infected mice, where there was a significant increase in M1 NK cells and decrease in M2 NK cells in the spleens and livers of TNFR1 and TNFR2 floxed mice (Supplementary Fig. 10f, g). Interestingly, these reductions in terminal maturation were also observed in naïve NK cells (Supplementary Fig. 11a), suggesting that TNF acts on NK cells during development and the changes were not entirely due to infection-induced TNF. Culture of NK cells in the presence of TNF induced minor increases in terminal maturation (Supplementary Fig. 11b), which was lost upon deletion of TNFR signaling (Supplementary Fig. 11c). These data suggest that TNF is important for the terminal maturation and proliferation of NK cells both in homeostasis and during infection.

## TNFRs 1 and 2 differentially modulate NK cell outcome

To assess which individual genes were different between genotypes, we tested for differential gene expression in pseudobulk replicates. Comparative analysis of gene expression between WT and conditional KO NK cells found 449 differentially expressed genes (DEGs) between WT and $Ncr1^{cre}Tnfr1^{fl/fl}$ NK cells (239 upregulated and 210 downregulated, $log_2FC > 0.5$ and adjusted P-value < 0.05 (Supplementary Data 1 and Fig. 6a)) and 568 between WT and $Ncr1^{cre}Tnfr2^{fl/fl}$ NK cells (278 upregulated and 290 downregulated (Supplementary Data 2 and Fig. 6b)). Of these DEGs, 155 were unique to $Ncr1^{cre}Tnfr1^{fl/fl}$, 274 were unique to $Ncr1^{cre}Tnfr2^{fl/fl}$, and 294 were shared between both (Supplementary Fig. 12a). Many of these shared DEGS were upregulation of genes associated with less mature NK cells (Cxcr3, Cd7, Tcf7, Cd27, Foxo3), as well as downregulation of effector genes (Gzmc, Gzmf) (Fig. 6a, b). Both conditional knockouts also had a downregulation of

Havcr2, encoding Tim3, further confirming that TNF drives Tim3 expression in our model (Fig. 6a, b). Of note, TNFR1 deficient NK cells also had a significant reduction in expression of apoptosis and cell death-associated genes (Uba52, Tmsb10, Lgals1, Anxa8, Larp1) as well as a significant upregulation of anti-apoptotic genes (Bcl2a1b, Klf4). Conversely, deletion of TNFR2 led to reductions in genes associated with proliferation/cell cycle (Hist1h1a, Hist1h3f, Mki67) as well as terminal maturation (Klrg1) (Fig. 6a, b). GO enrichment analysis of up and downregulated DEGs showed activation of pathways involved in various biological pathways and immune processes when either receptor was deleted (Supplementary Fig. 12b, c), however deletion of TNFR2 also led to suppression of various cell cycle and division-associated pathways (Supplementary Fig. 12c). When comparing the $log_2FC$ by deletion of either receptor, genes that were only downregulated in TNFR1-deficient cells were associated with apoptosis/cell death (Uba52, Tmsb10) and genes only downregulated in TNFR2-deficient cells associated with effector function (Tnfrsf25, Gzma) and proliferation (Mki67) (Supplementary Fig. 12d). There were few genes that were upregulated in one knockout and downregulated in the other, or vice versa, suggesting that deletion of one receptor did not increase the sensitivity of the other. Differential gene expression of each individual cluster found similar DEGs associated with apoptosis/cell death across each cluster when comparing WT to $Ncr1^{cre}Tnfr1^{fl/fl}$, reinforcing that DEGs in the bulk population were due to effects of TNFR1 signaling rather than changes in cluster proportions (Supplementary Figs. 13, 14). Conversely, differences in proliferation genes between WT and $Ncr1^{cre}Tnfr2^{fl/fl}$ were largely lost when comparing different clusters, likely due to these differences being represented by changes in the proportions of proliferating clusters (Supplementary Figs. 13, 14).

To investigate these mechanisms further, we performed gene set enrichment analysis (GSEA) of relevant pathways. We found a significant reduction in the hallmark pathways for Mtorc1 signaling, apoptosis, and E2F targets in both knockouts compared to the wild-type (Fig. 6c–e). $Ncr1^{cre}Tnfr1^{fl/fl}$ NK cells also had significantly reduced apoptosis signaling compared to $Tnfr2^{fl/fl}$ cells, further suggesting that apoptosis was a dominant pathway triggered by TNFR1. That E2F targets was reduced in both knockouts suggests that both TNFRs could promote proliferation. Together, these data are suggestive that TNFR1 and TNFR2 could play both overlapping and distinct roles, whereby both receptors promote proliferation and effector function in NK cells but TNFR1 simultaneously promotes cell death.

We used flow cytometry to confirm the scRNA-seq findings and investigate the potential for overlapping and opposing outcomes from TNFR1 and TNFR2 signaling. Mice were infected with S. Typhimurium, and spleens and livers taken at day four post-infection to investigate immune parameters and bacterial burdens. We found deletion of either TNFR1 or TNFR2 led to significant reductions in the expression of Tim3 in splenic NK cells (Fig. 6f). Curiously, we also saw TIGIT was upregulated after deleting each receptor (Fig. 6g), somewhat consistent with previous reports showing type I interferons increasing Tim3 expression but decreasing TIGIT in T cells[21]. When quantifying the total numbers of NK cells, we found deletion of TNFR1 increased the number of NK cells in the spleen compared to wild-type, whereas the deletion of TNFR2 reduced cell numbers (Fig. 6h), reinforcing the scRNA-seq data suggesting TNFR1 was leading to cell death/apoptosis and TNFR2 could contribute to proliferation/accumulation. However, no differences were observed in the liver (Fig. 6i). We used AnnexinV staining to further confirm that TNFR1, but not TNFR2, contributed to cell death during S. Typhimurium infection (Fig. 6j, k). Curiously, when examining proliferation by either EdU incorporation (Supplementary Fig. 15a, b) or Ki67 staining (Supplementary Fig. 15c, d), we found no differences between any groups. We also looked at levels of cytokines in the serum of infected mice and found deletion of TNFR2 ablated the

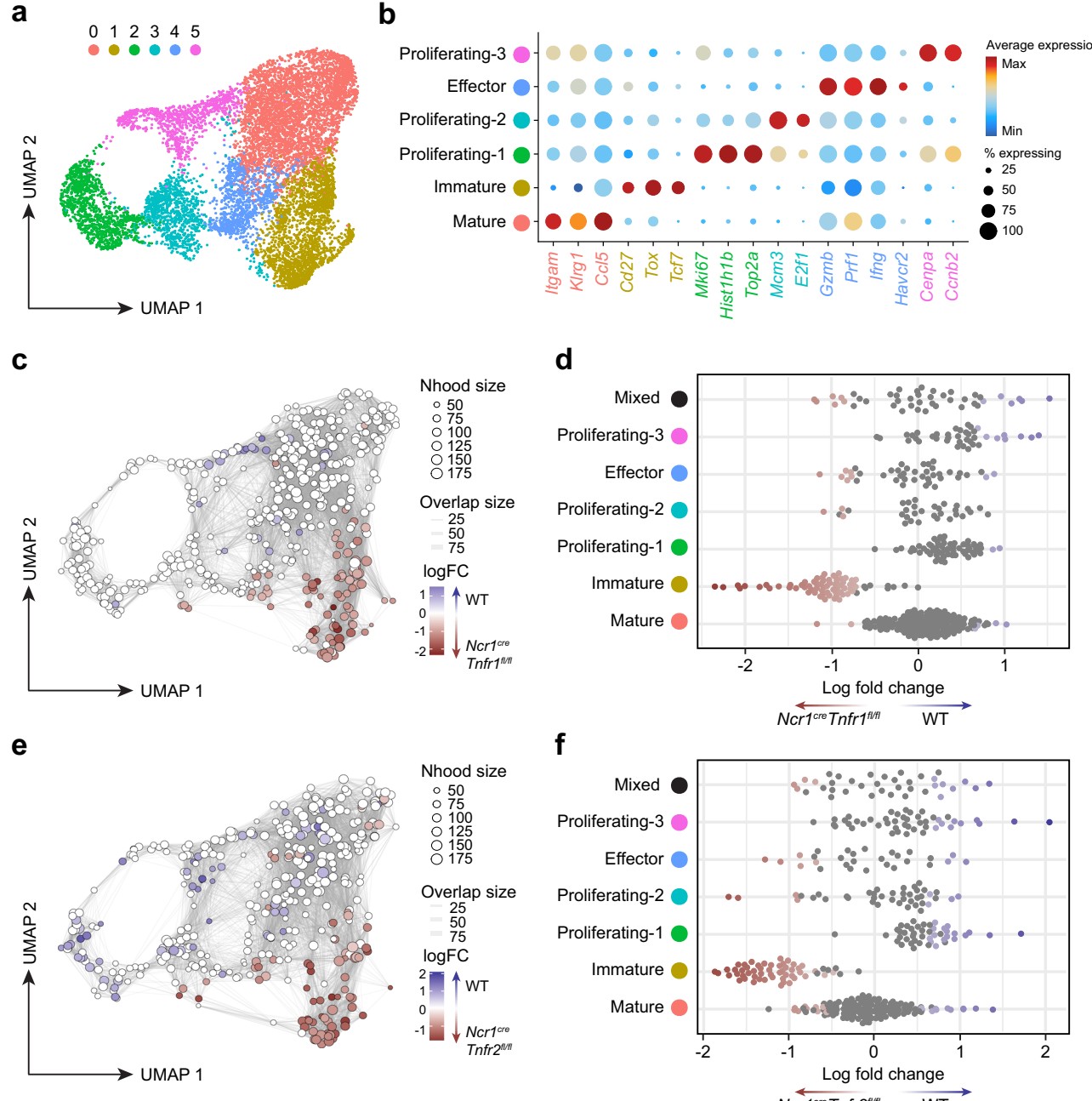

**Fig. 5 | Deletion of either TNFR1 or TNFR2 reduces NK cell terminal maturation.**
Transgenic *Ncr1^cre^*, *Ncr1^cre^Tnfr1^fl/fl^*, and *Ncr1^cre^Tnfr2^fl/fl^* mice were infected with *S.* Typhimurium, and NK cells analyzed by scRNA-seq on day four post-infection. **a** UMAP plot of NK cells colored by Seurat clusters, where each dot represents a single cell. **b** Dot plot showing the relative abundance of signature genes from each cluster. **c** UMAP plot of scRNA-seq differential abundance between WT and *Ncr1^cre^Tnfr1^fl/fl^* by Milo, where sampled neighborhoods are colored according to logFC. **d** Beeswarm plot showing the distribution of logFC between WT and

*Ncr1^cre^Tnfr1^fl/fl^* in neighborhoods separated by Seurat clusters. **e** UMAP plot of scRNA-seq differential abundance between WT and *Ncr1^cre^Tnfr2^fl/fl^* by Milo, where sampled neighborhoods are colored according to logFC. **f** Beeswarm plot showing the distribution of logFC between WT and *Ncr1^cre^Tnfr2^fl/fl^* in neighborhoods separated by Seurat clusters. For **c**–**f**, colored neighborhoods represent statistically significant differences (FDR < 0.05 by quasi-likelihood *F*-test). Data from a single experiment (*n* = 4 *Ncr1^cre^*, 4 *Ncr1^cre^Tnfr1^fl/fl^*, and 3 *Ncr1^cre^Tnfr2^fl/fl^*). Abbreviations: WT, wild-type; Nhood, neighborhood; logFC, log fold change.

titers of circulating IFN-γ (Fig. 6l). Lastly, to determine how these phenotypes might impact the ability of NK cells to contribute to immunity, we quantified bacterial burdens from each organ. We found that TNFR1 deficient mice had a reduced bacterial burden within the spleen (Fig. 6m), whereas TNFR2 deficient mice had increased bacterial burden in the liver (Fig. 6n). Taken together, these data shows that TNFR1 and TNFR2 have overlapping as well as distinct functions on the phenotype of NK cells and their ability to response to bacterial infection. TNFR1 primarily plays an inhibitory

role and prevents NK cell accumulation via cell death, whereas TNFR2 plays a protective role and promotes NK cell accumulation.

## TNF signaling is associated with NK dysfunction in human immune cells during bacterial infection
Thus far, this study has shown that TNF regulates NK cell function during bacterial infection in mice. We next investigated whether this could be the case in human disease by using a published scRNA-seq dataset from peripheral blood mononuclear cells

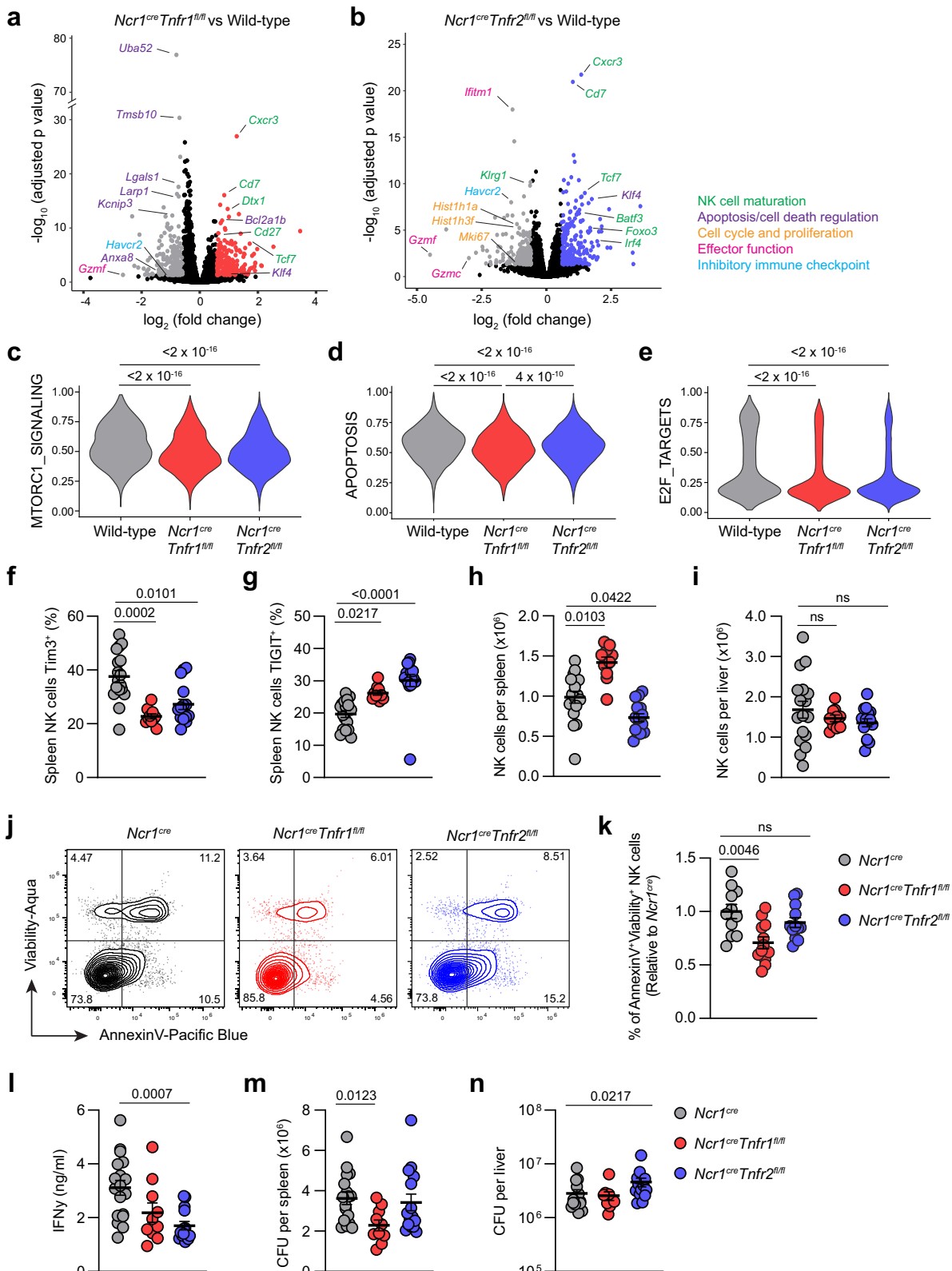

(PBMCs) of human donors pulsed with *S.* Typhimurium[31]. We could identify a population of NK cells in this dataset (Fig. 7a), which had high expression of *HAVCR2*, encoding Tim3 (Fig. 7b). Comparison of naive PBMC to those exposed to *S.* Typhimurium found that exposure to *S.* Typhimurium increased the expression of *HAVCR2* on NK cells (Fig. 7c). These data show that *S.* Typhimurium can induce expression of Tim3 on human NK cells, which

could potentially be through inducing TNF expression from other cell types.

A serious complication of bacterial infection is sepsis, which is characterized by high levels of inflammatory cytokines, including TNF. Thus, we predicted that the mechanisms of TNF-induced dysfunction could also be observed in human sepsis. To explore this, we used another previously published scRNA-seq dataset from PBMC of

**Fig. 6 | TNFR1 and TNFR2 have overlapping and opposing effects on NK cells.**
**a, b** Volcano plots showing differentially expressed genes between $Ncr1^{cre}Tnfr1^{fl/fl}$ and WT NK cells (**a**), or $Ncr1^{cre}Tnfr2^{fl/fl}$ and WT NK cells (**b**) from scRNA-seq. Gray dots represent genes upregulated in WT cells, red dots represent genes upregulated in $Ncr1^{cre}Tnfr1^{fl/fl}$ cells, and blue dots represent genes upregulated in $Ncr1^{cre}Tnfr2^{fl/fl}$ cells ($\log_2$FC > 0.5 and adjusted $P < 0.05$). **c–e** Violin plots showing relative enrichment of indicated Hallmark gene sets, calculated by GSEA. Transgenic $Ncr1^{cre}$, $Ncr1^{cre}Tnfr1^{fl/fl}$, and $Ncr1^{cre}Tnfr2^{fl/fl}$ mice were infected with *S.* Typhimurium, and spleens and livers analyzed at day four post-infection. **f** Expression of Tim3 on splenic NK cells. **g** Expression of TIGIT on spleen NK cells. **h, i** Total NK cell counts from spleen (**h**) and liver (**i**). **j** Representative flow cytometry plots showing AnnexinV and Viability staining. **k** Percentages of apoptotic NK cells (defined as AnnexinV$^+$Viability$^+$ and displayed as relative to $Ncr1^{cre}$ to normalize between experiments). **l** IFN-γ titers from the serum of infected mice at day four post-infection. **m, n** Bacterial burden per spleen (**m**) or liver (**n**), determined by CFU counts. Data from (**a–e**) are from a single experiment ($n = 4$ $Ncr1^{cre}$, 4 $Ncr1^{cre}Tnfr1^{fl/fl}$, and 3 $Ncr1^{cre}Tnfr2^{fl/fl}$), data from (**f–n**) are from two independent experiments ($n = 17$ $Ncr1^{cre}$, 10 $Ncr1^{cre}Tnfr1^{fl/fl}$, and 15 $Ncr1^{cre}Tnfr2^{fl/fl}$). Each dot represents one animal, and error bars indicate mean ± SEM. Groups were compared using the Wald test with Benjamini and Hochberg adjustment for (**a, b**), the two-sided Wilcoxon rank-sum test for (**c–e**), and Kruskal-Wallis test with Dunn's multiple comparisons test for (**f–n**), where $P < 0.05$ was considered statistically significant. Abbreviations: IFN, interferon; CFU, colony forming units. Source data are provided as a Source Data file.

patients with bacterial urinary tract infections and/or sepsis[32]. This dataset included healthy controls (Control), as well as patients with urinary tract infections (UTI) with leukocytosis but no signs of sepsis (Leuk-UTI), UTI with mild sepsis (Int-URO), and UTI with clear or persistent sepsis (URO). The dataset also included patients admitted to the intensive care unit (ICU) with or without sepsis (ICU-SEP and ICU-NoSEP, respectively), and patients with bacteremia and sepsis from hospital wards (Bac-SEP). Using the cell-type annotations in the original publication to identify the various immune cell populations (Fig. 7d), we isolated the NK cells and found they could be split into two clusters identifying less mature and mature NK cells (Fig. 7e). Expression of *HAVCR2* was observed predominantly in the mature NK cell cluster (Fig. 7f). Notably, the percentage of NK cells expressing *HAVCR2* was increased in some septic cohorts (Int-URO, URO, ICU-SEP) compared to non-septic (Leuk-UTI, ICU-NoSEP) (Fig. 7g). Other immune checkpoints, such as TIGIT, were unchanged between groups (Fig. 7h), consistent with our data using mouse models. Examination of *TNFRSF1A* and *TNFRSF1B*, encoding TNFR1 and TNFR2 respectively, found that while the expression levels were unchanged (Supplementary Fig. 16a, b), the percentage of NK cells expressing *TNFRSF1A* was increased in the URO group (Fig. 7i), whereas the percentage of NK cells expressing *TNFRSF1B* was increased in the Int-URO and URO groups (Fig. 7j), further supporting a potential role for TNF signaling. Analysis of genes associated with the Hallmark gene sets for TNF signaling and apoptosis identified that many genes were more abundant in the septic cases compared to the non-septic controls (Fig. 7k), suggesting that this upregulation of Tim3 could be a biomarker of TNF-induced dysfunction and apoptosis in human NK cells during sepsis. Together, these results suggest that Tim3 expression and TNF signaling may also be relevant for NK cells in human diseases, including sepsis.

## Discussion

Cytokines can have a wide range of effects on NK cell function, and impact the ability of these innate immune cells to provide host immunity against viral, fungal, and bacterial pathogens. Previously, the effects of TNF on NK cell function have been poorly defined. In this study, we have found TNF signaling is a major regulator of NK cell function during bacterial infection. Tim3, an inhibitory checkpoint molecule expressed on the surface of NK cells, was found to be a biomarker of TNF signaling in NK cells. Using transgenic mouse models of Tim3-overexpression in NK cells, we show that in certain circumstances, Tim3 can impair NK cell function and survival. It is important to note that this model induced supraphysiological levels of Tim3, and thus may not reflect natural functions of Tim3 during infection. Interestingly, the literature is enigmatic when describing Tim3 expression in NK cells. Tim3 has been reported as an inhibitory receptor that can impede cytotoxic responses of NK cells during pregnancy[33] and against tumors[34], without suppressing IFN-γ[35]. Other studies have suggested that Tim3 is an important stimulatory receptor to promote IFN-γ production[36]. However, in models of LPS-induced endotoxemia, Tim3 blockade was able to enhance IFN-γ production in NK cells[37]. These conflicting results may arise from the inability of anti-Tim3 antibodies to fully block all ligands of Tim3[38], or the NK cell effects being secondary to anti-Tim3 blockade working on other cell types, such as DCs[39]. Regardless, at physiological levels Tim3 expression did not appear to induce NK cell dysfunction in our models, however it did act as a biomarker for TNF signaling. More studies are required to fully appreciate how this TNF-Tim3 axis might impact NK cell functionality across different disease states.

We used Tim3 expression as a clue to show that TNF significantly impacts NK cells and their ability to respond to bacterial pathogens. TNF exerts its effects on NK cells by acting upon both TNFR1 and TNFR2, which have overlapping and distinct actions on NK cell function. Of note, TNF was able to upregulate the immune checkpoint Tim3 via both TNFR1 and TNFR2. While anti-TNF therapies have long been a mainstay in the treatment of auto-immune diseases such as rheumatoid arthritis, they are increasingly being investigated in cancers. In particular anti-TNF has been suggested as an adjunct to cancer immunotherapies, such as immune checkpoint blockade (ICB)[40]. Not only can anti-TNF therapy improve the anti-tumor efficacy of ICB, but it also ameliorates immune-related adverse events such as colitis[41]. Clinical trials have attempted to neutralize TNF in the context of sepsis with mixed results, however, a meta-analysis suggested that this type of therapy could reduce overall mortality[42].

A major finding from our study was the dichotomous outcomes of TNFR1 and TNFR2 in NK cells, where TNFR1 led to NK cell death and reduced accumulation while TNFR2 promoted accumulation. Importantly, one aspect not investigated in this study is NK cell migration/trafficking, which could be playing an important role in the differences in NK cell accumulation in our various mouse models. Addressing this would require transgenic mouse models such as the photoconverting Kaede model[43] or models of homing receptor deficiency, which we did not have access to for the purpose of this study. Regardless, we have previously observed an organ specific NK cell loss following *S.* Typhimurium infection[17], which the current study suggests can be attributed in part to TNFR1 signaling inducing cell death. We have also shown that TNF can contribute to NK cell proliferation, but our results are somewhat conflicting as to the relative contribution of each receptor. Our scRNA-seq analysis suggests TNFR2 was the dominant receptor contributing to proliferation, however this finding could not be replicated by flow cytometry. Due to the shared downstream signaling pathways of each receptor, it is also likely there is significant overlap between each receptor, whereby each contribute to proliferation to some degree. This is supported by our GSEA results which found deletion of TNFR1 or TNFR2 had impacts on genes associated with apoptosis and E2F signaling (predicting proliferation). However, we observed stark differences in overall NK cell numbers upon deletion of either receptor. This dichotomy is likely due to subtle differences in the downstream signaling of TNFR1 and TNFR2. In particular, a primary difference is that TNFR1 contains a death domain which is able to trigger cell death via activation of caspases[44]. These differences in signaling between TNFR1 and TNFR2 allowed each receptor to have a different outcome when acting on NK cells. Individual roles for TNFR1

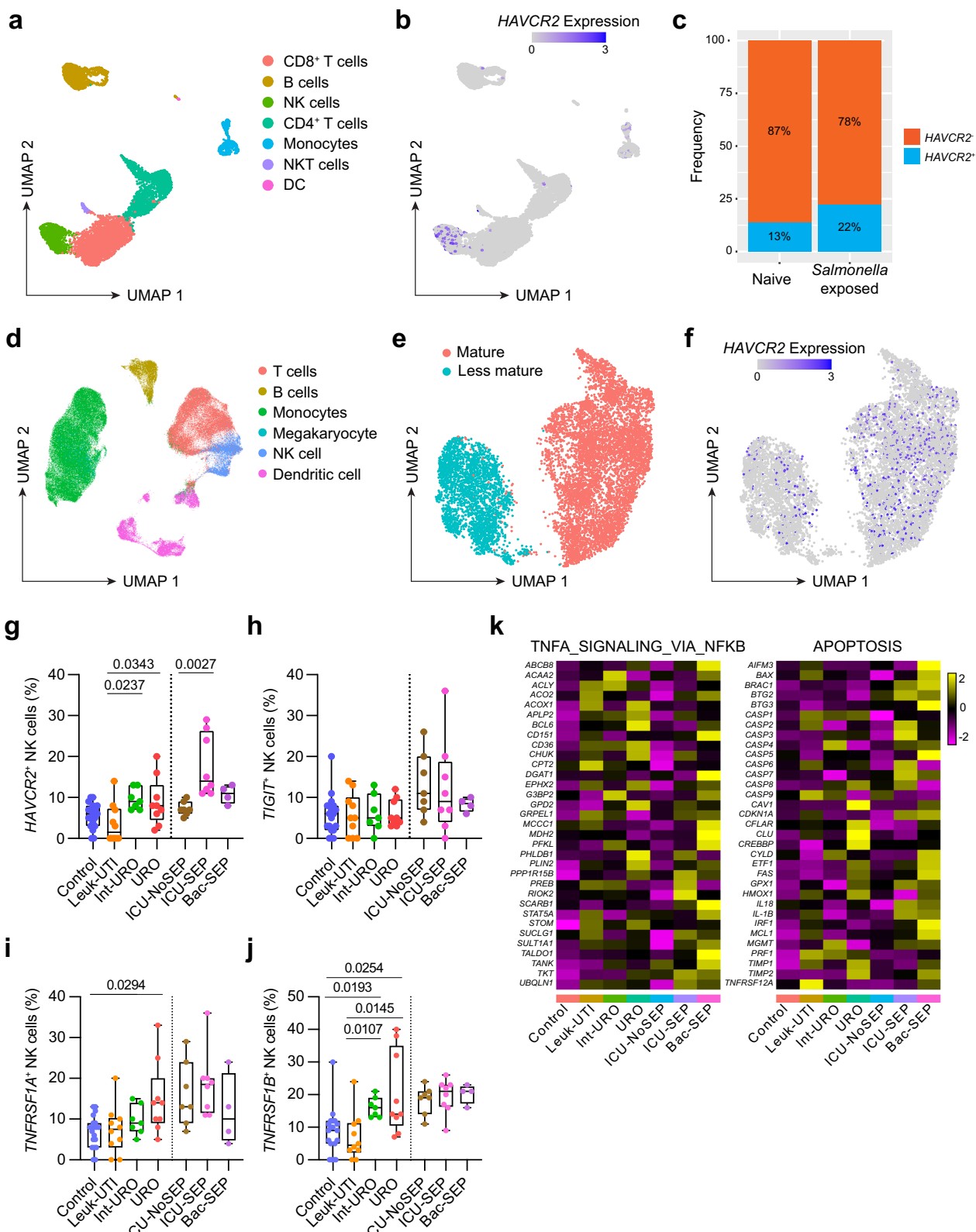

and TNFR2 have also been observed in other immune cells. As an example, in CD4[+] T cells, these two receptors oppose each other in the maintenance of autoimmunity. TNFR1 promotes the differentiation of inflammatory subsets of T cells, including Th1 and Th17, to exacerbate autoimmunity, whereas TNFR2 promotes the differentiation and function of protective Tregs[13]. Curiously, CD8[+] T cells appear different to our results in NK cells, whereby TNFR2 induces cell death through

overactivation, and TNFR1 limits this response to promote survival[45]. In contrast, in myeloid cells such as macrophages, TNF acting through TNFR1 is a well-characterized trigger of apoptosis and necroptosis[46]. However, TNFR2 can also sensitize macrophages towards being receptive to necroptosis in response to TNFR1 signaling[47]. In neutrophils, both TNFR's promote neutrophil-driven clearance of bacterial skin infection, where TNFR1 is responsible for the trafficking of

**Fig. 7 | scRNA-seq reveals Tim3 and TNF signaling in NK cells from human PBMC.** Reanalysis of scRNA-seq from PBMC pulsed with *S.* Typhimurium. **a** UMAP showing total cells from PBMC cultures. **b** UMAP showing *HAVCR2* expression in PBMC. **c** Relative proportions of *HAVCR2-* or *HAVCR2 +* NK cells from naïve or *S.* Typhimurium pulsed PBMC. Reanalysis of scRNA-seq of PBMC from sepsis patients. **d** UMAP showing total cells from PBMC of sepsis patients and healthy controls, divided cell type. **e** NK cells divided up the Seurat cluster. **f** UMAP showing *HAVCR2* expression in NK cells. **g**–**j** Box plots showing the percentage of NK cells expressing either *HAVCR2*, *TIGIT*, *TNFRSF1A*, or *TNFRSF1B*. Box represents the upper and lower quartile, and the median and whiskers show range between minima and maxima. Vertical dotted line separates cohorts from different hospitals. **k** Heatmap showing average expression in NK cells of selected genes from the indicated HALLMARK genesets. All data from published datasets (**g**–**k**: *n* = 19 control, 10 Leuk-UTI, 7 Int-URO, 9 URO, 7 ICU-NoSEP, 8 ICU-SEP, and 4 Bac-SEP). Groups were compared using the Kruskal-Wallis test with Dunn's multiple comparisons test, where *P* < 0.05 was considered statistically significant. Abbreviations: UTI, urinary tract infection; URO, urosepsis; ICU, intensive care unit; SEP, sepsis; Bac, bacterial. Source data are provided as a Source Data file.

neutrophils to the site of infection, but TNFR2 is responsible for direct antibacterial function[48]. Taken together, TNFR1 and TNFR2 clearly play different roles in different immune cells, and more work is required to tease apart the relative contribution of each receptor in various cell types in different inflammatory settings.

Interestingly, our results show that deletion of TNFR1 on NK cells improved bacterial burdens in the spleen, whereas deletion of TNFR2 worsened bacterial burdens in the liver. The strain of *S.* Typhimurium used in this study (SL3261) induces a chronic infection which is sublethal in mice, and so animal survival has not been scored. However, we believe the differences in NK cell function and bacterial burden we show strongly demonstrate the effects of the TNFRs on NK cells in our model. An interesting future experiment could be to explore the roles of the TNFRs on NK cells in an acute/lethal model, to determine how the differences in organ burden translate to animal survival. Why these two receptors would be playing roles in different organs remains unclear, however this could be due to differences in the presentation of TNF between the two organs. As TNFR1 can be activated by both soluble and membrane-bound TNF, whereas TNFR2 can only be activated by membrane-bound[49], our results may suggest an emphasis on soluble TNF in the spleen allowing the TNFR1 response to dominate. Regardless, the dichotomy between the two receptors raises important questions on how the regulation of NK cells in response to TNFR1 or TNFR2 can be targeted, and whether targeting specific receptors, rather than TNF itself, may be advantageous in certain pathologies. For example, a TNFR1-neutralizing antibody may provide the various benefits of anti-TNF, while leaving the TNF-TNFR2 interaction functional. This has been trialed previously in models of arthritis, where TNFR1 blockade provided the same benefits as neutralization of TNF[50]. This idea has also been raised in mouse models of sepsis, as TNFR1[-/-] mice had prolonged survival and TNFR2[-/-] mice reduced survival compared to WT controls in response to cecal ligation puncture[51], further giving weight to targeting TNFR1. NK cells have been shown to deplete early within the blood of sepsis patients[52], and this depletion is sustained for at least 14 days[53]. Our study suggests TNF-TNFR1 signaling may be a primary driver of this NK cell depletion through induction of cell death.

In cancers, whether targeting the TNFR1 pathway would be viable remains unknown, although it should be noted that TNFR2 promotes the suppressive function of regulatory T cells[13,54], so while preserving TNFR2 over TNFR1 signaling may improve NK cell responses, it may also promote immunosuppression. Targeting the inhibitory TNFR1 pathway may be more efficacious for cellular therapies. Following the success of the adoptive transfer of T cell products, NK cells are gaining attention for their potential as cellular therapies[55]. Importantly, NK cells can be genetically modified to delete inhibitory receptors that impede their function. For example, NK cells have been edited to delete inhibitory molecules TGFBRII (receptor for TGF-β) or cytokine-inducible SH2-containing protein (CIS) in order to enhance their anti-tumor efficacy[56–58]. This has the added benefit of only targeting the adoptively transferred cells, preventing the off-target toxicities often associated with targeting inhibitory molecules systemically. Our study suggests that deletion of TNFR1 may also improve the responsiveness of NK cellular therapies, and this should be examined accordingly.

In summary, we found Tim3 to be a biomarker of TNF signaling in NK cells, and this signaling pathway as a major regulator of NK cell-mediated immunity during bacterial infection. While the clinical use of NK cell-based therapies for the treatment of infection and/or sepsis are still in development, our findings suggest TNF signaling is a highly valuable pathway and specific targeting of TNFR1 or TNFR2 could be exploited in the future of NK cell-based immunotherapies. This pathway may also be highly relevant in other diseases, such as cancer, and thus we propose that this mechanism be investigated in other settings to assess how our findings could apply to a wider array of disease states.

## Methods

### Mice

*Ncr1[cre]* mice[59] were crossed with *Tim3[fl/fl]* mice[60] or *FSF-Tim3* mice[61] to generate novel strains with condition deletion (*Ncr1[cre]Tim3[fl/fl]*) or over-expression (*Ncr1[cre]FSF-Tim3*) of Tim3 specifically within NK cells. *Ncr1[cre]* mice were also crossed with *Tnfr1[fl/fl]* and/or *Tnfr2[fl/fl]* mice[62] to generate novel strains with conditional deletion of TNFR1 (*Ncr1[cre]Tnfr1[fl/fl]*), TNFR2 (*Ncr1[cre]Tnfr2[fl/fl]*), or both TNFRs (*Ncr1[cre]Tnfr1[fl/fl]Tnfr2[fl/fl]*) specifically within NK cells. *Ncr1[cre/wt]* were used as wild-type controls for transgenic mouse experiments. *Rag2[-/-]γc[-/-]* mice (JAX strain number 014593) were purchased from The Walter and Eliza Hall Institute for Medical Research (WEHI) and used as lymphocyte-deficient controls for adoptive transfer experiments. All transgenic mice were developed on the C57BL/6 background and bred and maintained at the Translational Research Institute (TRI) Biological Resources Facility (BRF). C57BL/6 mice (JAX strain number 000664) were purchased from Ozgene Animal Resources Center (ARC) and used as wild-type mice in vitro screenings. Mice were bred and housed at the TRI BRF in specific pathogen-free conditions in ventilated cages (22 °C and 50–70% relative humidity) under a 12-hour light cycle. All animal experiments were conducted according to approval by the University of Queensland Health Science Animal Ethics Committee (approval numbers UQDI/536/19, 2021/AE000584, and 2021/AE000585), and procedures were carried out in accordance with the regulatory standards of the National Health and Medical Research Council (NHMRC) and the Australian Code for the Responsible Conduct of Research. Carbon dioxide asphyxiation was used to euthanize experimental mice. Mice were to be euthanized early if they lost greater than 20% bodyweight over the course of an experiment, although this criteria was not required for this study. All experiments were performed using 8–16 week-old age and sex-matched mice. Both male and female mice were used for experiments and randomly distributed among groups.

### Bacterial strains and in vivo infections

Mice were infected with an attenuated *aroA* mutant strain of *Salmonella enterica* subspecies Typhimurium SL3261, or wild-type SL1344[18]. For in vivo infection, bacteria were grown at 37 °C with shaking in Lysogeny broth (LB) for 16 to 18 h. $OD_{600}$ was used to enumerate bacteria, before being diluted to the appropriate concentration in PBS. Mice received SL3261 at $5 \times 10^6$ CFU in 200 μl by intraperitoneal (*i.p.*) injection to induce systemic infection, or SL1344 at $1 \times 10^8$ CFU in 100 μl by oral gavage.

## In vivo mouse treatments

Anti-NK1.1 (mIgG2a, clone PK136, BE0036, BioXCell) was used to deplete NK cells and was given *i.p.* at a dose of 100 μg per mouse in 200 μl of PBS three days before infection, on the same day as infection, and then every five days for the remainder of the experiment[63]. Anti-Tim3 (rat IgG2a, clone RMT3-23, BE0115, BioXCell) or isotype control (rat IgG2a, clone 2A3, BE0089, BioXCell) were given as *i.p.* injections of 200 μg in 200 μl of PBS 24 h post-infection and every three days thereafter, as previously described[64].

## Mouse tissue processing

Serum samples were taken from mice by isolating blood by retro-orbital bleeds, centrifuging at $2000 \times g$ for 10 min, and removing serum from the cell pellet. Serum samples were stored at − 20 °C until analysis. At experimental endpoints, mice were euthanized by $CO_2$ asphyxiation, and spleens and/or livers harvested for processing for flow cytometry and bacterial burdens. Bacterial burdens were quantified from organs by homogenizing samples in PBS with 0.1% Triton-X (X100-100ML, Sigma-Aldrich) before serially diluting in PBS and plating on LB agar.

## Flow cytometry

Spleens and livers were harvested and single-cell suspensions generating by passing organ through 70 μm or 100 μm cell strainers, respectively. Liver immune cells were further enriched using 37.5% Percoll (17089101, Cytiva Life Sciences). Red blood cells (RBC) were lysed by incubation in RBC Lysis Buffer (420302, Biolegend). Dead cells were stained by incubation in Fixable Viability Stain 440UV (BD Bioscience). Fc receptors were blocked in FcR Blocking Reagent (130-092-575, Miltenyi Biotec) for 15 min before staining of surface antigens was performed for 45 min at room temperature (Supplementary Table 1). A lineage cocktail of biotinylated anti-CD19, anti-F4/80, and anti-Ly6G was used for B cell, macrophage, and neutrophil exclusion. For AnnexinV staining, cells were stained for 15 min at room temperature with AnnexinV-Pacific Blue (640918, Biolegend) in AnnexinV binding buffer (422201, Biolegend). For intracellular antigen staining, cells were fixed and permeabilized using the FoxP3/Transcription Factor Staining Buffer Set (00-5523-00, eBioscience), then stained with intracellular antibodies for one hour at room temperature (Supplementary Table 1). Liquid Counting Beads (335925, BD Biosciences) were added to enumerate cell numbers. Data was acquired on a BD FACSymphony A5 (BD Biosciences) or a Cytek Aurora (Cytek). Flow cytometry data analysis was performed using FlowJo software (v10.9, Treestar).

## EdU incorporation assay

5-ethynyl-2'deoxyuridine (EdU) incorporation was performed as previously described[65], using a Click-IT Plus EdU AF488 Flow Cytometry Assay Kit (C10632, Thermo Fisher Scientific). Briefly, mice were injected with 200 μg of EdU on days 1 and 3 post-infection. On day 4 post-infection, spleens were processed as described and stained for EdU according to the manufacturer's instructions.

## Adoptive NK transfer

NK cells were isolated from spleens of *Ncr1^cre^Tim3^fl/fl^* and *Ncr1^cre^FSF-Tim3* mice using an NK Cell Isolation kit (130-115-818, Miltenyi) as per manufacturer's instructions. Freshly sorted NK cells were mixed 50:50, and $5 \times 10^5$ total cells injected intravenously into recipient *Rag2^-/-^γc^-/-^* mice. After seven days, mice were either infected with $1 \times 10^6$ *S.* Typhimurium SL3261 by *i.p.* injection or left naïve. Seven days post-infection, relative NK cell numbers were assessed in spleens and livers by flow cytometry.

## In vitro NK cell culture

For in vitro studies, NK cells were harvested from mouse spleens. Spleens processed into single-cell suspensions through 70 μm cell strainers and NK cells enriched by negative selection using an NK Cell Isolation kit (130-115-818, Miltenyi) as per manufacturer's instructions. NK cells were stained with Cell Trace Violet (CTV, C34557, Invitrogen), then seeded into 96-well round bottom plates at 15,000 cells per well in 200 μl of IMDM (SH30228.01, Cytiva Life Sciences) supplemented with 10% FCS, 1% GlutaMAX (35050061, Gibco), 0.1% 2-mercaptoethanol (M3148, Sigma-Aldrich), and 1% penicillin-streptomycin (P4333, Sigma-Aldrich). Cultures included recombinant cytokines, including 10 ng/ml human IL-15 (200-15, Peprotech), and the indicated concentrations of murine TNF (315-01 A, Peprotech), murine IFN-α (14-8312-80, eBioscience), or murine TGF-β (14-8342-80, eBioscience). In some experiments, NK cells were cultured in the presence of TNF pathway inhibitors BI605906 (19184, Cayman Chemical Company), SP600125 (10010466, Cayman Chemical Company), Lomapimod (SML3596, Sigma Aldrich), Amgen16 (SML2457, Sigma Aldrich), or Rapamycin (13346, Cayman Chemical Company). Plates were incubated at 37 °C with 5% $CO_2$ for three days before processing for flow cytometry.

NK cell survival and proliferation were determined by calculating the cohort number and mean division number, respectively, as previously described[28,66]. CTV peaks were given a generation number, $i$, where undivided cells were given $i = 0$ and increasing by 1 for each subsequent peak. The total cohort number was then determined to estimate NK cell death by accounting for the confounding influence of proliferation. To determine the total cohort number the number of each of cells within generation $i$ was divided by $2^i$ to generate the cohort number for generation $i$. The total cohort number was then calculated by summing the cohort numbers for each generation:

$$Total\ cohort\ number = \sum \frac{cell\ number_i}{2^i}$$

Each biological replicate was then normalized to its corresponding IL-15 only control to obtain a relative survival value.

NK cell mean division number was determined by calculating the average number of divisions in each condition. Each generation number $i$ is multiplied by the fraction of the total cohort which has undergone $i$ divisions. The value of each generation was then summed to generate the mean division number:

$$Mean\ division\ number = \sum \left( i \times \frac{cohort\ number_i}{total\ cohort\ number} \right)$$

## Splenocyte stimulation

ELISA plates were coated at 4 °C for 24 h with 80 μl per well of anti-NK1.1 antibody (BE0036, BioXCell, 10 μg/ml in carbonate bicarbonate buffer). Antibody was removed, wells washed twice with PBS, then blocked with 100 μl of IMDM + 10% FCS at 37 °C for at least 10 min. $1 \times 10^6$ splenocytes were cultured in IMDM with 10% FCS in the presence of Monensin (420701, Biolegend), Brefeldin A (420601, Biolegend), anti-CD107a (BD Biosciences), and the indicated cytokines (human IL-15 at 25 ng/ml (200-15, Peprotech), murine IL-12 at 25 ng/ml (210-12, Peprotech), and/or murine IL-18 at 5 ng/ml (PMC0184, Gibco)) for four hours at 37 °C.

## Measurement of cytokines

IFN-γ titers were determined from murine plasma samples using a Mouse IFN-γ ELISA set (551866, BD Biosciences) as per the manufacturer's instructions. TNF titers were determined from murine plasma samples using a CBA Mouse TNF Flex Set (558299, BD Biosciences) as per the manufacturer's instructions.

### Single-cell RNA-sequencing (scRNA-seq)

NK cells were sorted as Lineage⁻NKp46⁺NK1.1⁺ from the spleens of naïve *Ncr1^cre^*, and day 4 infected *Ncr1^cre^*, *Ncr1^cre^Tnfr1^fl/fl^*, or *Ncr1^cre^Tnfr2^fl/fl^* mice. Biological replicates were stained with TotalSeq-A Hashtag oligos (HTO, Cat#'s 155801, 155803, 155805, and 155807, Biolegend) for multiplexing. Biological replicates were pooled, and single cells encapsulated into droplets using a Chromium Controller (10X Genomics) as per the manufacturer's instructions. Libraries were prepared using the Chromium Next GEM Single Cell 3' Kit v3.1 (1000269, 10x Genomics) as per the manufacturer's instructions, with additional steps to amplify the cell hashtag libraries, as per the Biolegend instructions. Libraries were sequenced using a NovaSeq6000 SP SE100 flow cell (Illumina).

### scRNA-seq computational methods

Reads from scRNA-seq experiments were aligned to the GRCm38 (mm10) mouse reference genome, quantified and demultiplexed into Gene Expression and Multiplexing Capture using cellranger multi (v7.1.0, 10x Genomics). Further analysis was performed using the Seurat package (v4.3.0)[67] using R. Outliers were excluded based on number of genes detected (<1000, >6500) or frequency of mitochondrial RNA transcripts (>5%). HTO counts were added to the scRNA-seq object, then the center log ratio normalized and samples demultiplexed using HTOdemux. Doublets were excluded based on the expression of multiple HTO reads, and contaminating ILC1 excluded based on the expression of *Cd200r1* and *Tnfsf10* and lack of expression of *Eomes*. Counts were normalized and the 2000 most variable genes were used as integration anchors to merge datasets from each genotype. The integrated sample counts were scaled, and variable features used for principal component analysis (PCA). The top 15 principal components from this analysis were then used as an input for dimensionality reduction by UMAP. Shared-nearest-neighbor based clustering using the top 15 principal components was used to generate clusters with a resolution = 0.3. The signature genes identifying each cluster were found using FindAllMarkers. The CellCycleScoring function of Seurat was used for cell cycle analysis. Differential abundance of cellular neighborhoods was performed in MiloR (v1.2.0)[30]. Gene set enrichment analysis (GSEA) was performed using the escape package (v1.8.0)[68]. Differential expression was determined by creating pseudobulk-replicates and using DESeq2[69] to identify differentially expressed genes.

### Statistical analysis

All graphs show mean ± SEM. Statistical analyses were performed in GraphPad Prism 9 or R. Statistical tests were applied as indicated in figure legends and significant differences marked on all figures.

### Reporting summary

Further information on research design is available in the Nature Portfolio Reporting Summary linked to this article.

## Data availability

The scRNA-seq datasets generated from this study have been deposited in the GEO repository database under the accession number GSE233790. Published datasets from Ben-Moshe et al.[31] can be accessed from the GEO repository database under the accession number GSE122084, and the dataset from Reyes et al.[32] can be accessed from the Broad Institute Single Cell Portal under number SCP548. All data are included in the Supplementary Information or available from the authors, as are unique reagents used in this Article. The raw numbers for charts and graphs are available in the Source Data file whenever possible. Source data are provided in this paper.

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

## Acknowledgements

We thank members of the Guimaraes and Wells laboratories, Prof. G. Belz, Prof. S. Bell, Prof. M. Sweet, Dr. C. Antilla, Dr. D. Dlugolenski, G. McDermott, Prof. J. Sun, and Dr. A. Mujal for discussion, comments, and advice in this project; the Translational Research Institute Flow Cytometry lab for flow cytometry services; Dr. V. Coyne and the Central Analytical Research Facility at Queensland University of Technology for sequencing services; Prof. H. Wajant for providing reagents; and Prof. E. Vivier for sharing the *Ncr1*$^{cre}$ mice. The generation of *Tnfr1*$^{fl/fl}$ and *Tnfr2*$^{fl/fl}$ mice used in this study was supported by Phenomics Australia and the Australian Government through the National Collaborative Research Infrastructure Strategy (NCRIS) program. This research was carried out at the Translational Research Institute, Woolloongabba, QLD 4102, Australia. The Translational Research Institute is supported by grants from the Australian and Queensland Governments. The Guimaraes Laboratory is funded by a US Department of Defense – Breast Cancer Research Program – Breakthrough Award Level 1 (#BC200025), and a grant (#2019485) awarded through the Medical Research Future Fund in a co-funded partnership with the QLD Children's Hospital Foundation, Microba Life Sciences, Miltenyi Biotec, the Richie's Rainbow Foundation, the Translational Research Institute, and UQ, a grant from the National Breast Cancer Foundation, Australia under award number 2023/IIRS0063, and funding from the Cooper Rice-Brading Foundation, Australia; The Tie Dye Project, Australia; The Kids Cancer Project, Australia; Tour de Cure, Australia; the PA Research Foundation, Australia. T.R.M., L.F.A., E.C., and J.K.M.W. were funded by the Australian Government Research Training Program Scholarship. C.K., F.S.F.G., and L.A. are funded by a Metro South Health Research Support Scheme – Co-Funded Collaboration Grant (#RSS_2023_085).

## Author contributions

T.R.M. and F.S.F.G. designed the research and wrote the paper. T.R.M., G.R.R., L.A., E.C., P.Y.L, J.K.M.W., S.K., T.J.W., and F.S.F.G. performed research and T.R.M., G.R.R, and L.A. analyzed data. M.H., A.J.K., P.A.K., C.W., C.K., and L.K. provided critical mouse models and advice in assays. T.J.W. and F.S.F.G. supervised work. This work was funded by grants led by F.S.F.G.

## Funding

## Competing interests

The authors declare no competing interests.
