## [Transparent Peer Review file · Nature Communications]

Dichotomous outcomes of TNFR1 and TNFR2 signaling in NK cell-mediated immune responses during inflammation

Corresponding Author: Dr Timothy McCulloch

Version 0:

Reviewer comments:

Reviewer #1

(Remarks to the Author)

This manuscript presents a study investigating the role of tumor necrosis factor (TNF) in regulating natural killer (NK) cells during bacterial infection. The researchers identified Tim3 as a biomarker of TNF signaling in NK cells during infection. Mouse models were used to explore the functions of TNF receptors (TNFR1 and TNFR2) in NK cells during bacterial infection. Deletion of TNFR1 enhanced cell survival and immunity, while TNFR2 deletion limited NK cell function and worsened immunity. The study also observed similar patterns in NK cells from human sepsis.

However, there are several issues noted in the manuscript:

The effect of *S. Typhimurium* infection on NK cell differentiation, proliferation, and migration in the liver should be clarified. The elevated expression of Tim3 might be related to increased NK cell presence. Fig 1f needs clarification regarding the y-axis, whether it represents Tim3-positive NK cells in the total NK population or total spleen cells.

The impact of TNFR1 and TNFR2 deletion on the total NK cell population in the liver and spleen before and after infection needs clarification. Fig 3c shows a reduction in NK cell numbers, which contradicts the statement in the text that "there was no difference in NK cell numbers in the TNFR floxed mice."

The conclusion that TNF is important for NK cell maturation, differentiation, and proliferation should be supported by data obtained from non-infected mice as well, as all data were collected from mice infected with bacteria.

The mechanism by which TNF induces Tim3 and its potential effects on Tim4 should be examined or discussed. The authors suggest that TNFR1 or TNFR2 knockout down-regulates Tim3, and the underlying mechanisms for this regulation should be explored.

The differing effects of TNFR deletion observed in the spleen and liver, as shown in Fig 5e, should be discussed.

The figure legend in Fig 5 does not match with the panel label, requiring correction.

The authors propose that TNFR1 and TNFR2 regulate NK cell death/apoptosis and proliferation, respectively. It would be interesting to explore the expression pattern of TNFR1 and TNFR2 in different NK cell clusters using scRNA-seq data.

In Fig 6, it would be beneficial to compare the expression levels of TNFR1 and TNFR2 in NK cells of septic patients versus non-septic patients.

The distinct or opposite effects of TNFR1 and TNFR2 on other immune cells, such as macrophages, should be compared and discussed.

Minor issues, including typos and missing words, should be carefully reviewed and corrected.

Reviewer #2

(Remarks to the Author)

The Authors have elucidated the role of TNFR 1/2 in NK cells in the differential outcome of the bacterial infection in mice model.

The authors have interesting findings and will benefit from looking into the physiological relevance of the TNFR1/2- Tim3 - axis.

TNFR1/2 brings about differential outcomes where TNFR1 deletion promoted cell survival and proliferation of NK cells and TNFR2 deletion limited cell proliferation

1. Percentage of NK cells expressing Tim3 is high post 2 days infection , which gradually falls. What is the status of TNFR1 and TNFR2 expression on the same subsets? What is the level of Serum TNF and IFN- γ at the same time period?
2. Survival Index of floxed mice (both TNFR1^{-/-} and 2^{-/-}, WT) must be scored to elucidate the final outcome of the infection process.
3. CFU analysis of bacterial burden in these floxed mice. Does the dissemination of Salmonella gets perturbed in either of the floxed mice?
4. In this particular article, Salmonella Typhimurium infection drives NK cell loss and conversion to ILC1-like cells, and CIS inhibition enhances antibacterial immunity, Timothy R. McCulloch, Gustavo R. Rossi, Timothy J. Wells, Fernando Souza-Fonseca Guimaraes doi:<https://doi.org/10.1101/2021.11.29.470332>, STM infection is reported to drive NK cell loss. One of the author Gustavo Rossi, is common in both the papers. It would be worthwhile to see that the loss of NK cells by STM is linked to TNFR1 and 2. In the floxed mice, the authors can look for the plasticity of NK cells.
5. Physiological route of infection by oral gavaging must be performed to elucidate the role of NK cells. One experiment with the percentage of NK cells with Tim3, TNFR 1/2 will suffice.
6. Organ burden and Survival index must be demonstrated to convincingly show the importance of the dichotomy in the disease outcome.
7. Though the human data-set for sepsis is good to know, it does not add anything to this manuscript. For this data to be meaningful, the isolation PBMC must be pulsed with STM and the NK cells markers and Tim3 expression can be followed.

Reviewer #3

(Remarks to the Author)

This study by McCulloch et al uses mice with NK cell-specific deficiencies in Tim3, TNFR1, and TNFR2 to explore the roles of these factors in NK cell maturation, function, and host defense against bacterial infection in vivo. The data suggest that Tim3 expression is regulated downstream of TNFR signaling NK cells, but deletion of Tim3 in NK cells has little effect on NK cell development/maturation at baseline or control of S. Typhimurium infection in vivo. The major finding of the study is that while both TNFR1 and TNFR2 regulate NK cell abundance and maturation in vivo, the individual receptors may have distinct and somewhat opposing roles. Specifically, TNFR1 may act to limit NK cell survival and anti-bacterial activity, whereas TNFR2 may act to promote NK cell proliferation. The discovery of dichotomous roles for TNFR1 and TNFR2 in NK cell biology is intriguing and novel, and is expected to be of interest to the field. Overall, the manuscript is well-written and the data are clearly presented. Nevertheless, addressing the following concerns would significantly improve the depth of the study and strengthen the key conclusions of the manuscript.

Major criticisms:

1. The main finding that TNFR1 and TNFR2 have differential roles in regulating NK cell survival and proliferation, respectively, is suggested by the data but not rigorously verified. Indeed, the study generally lacks experiments and assays designed to specifically measure cellular proliferation (CTV dye dilution, BrDU incorporation assays), cell death (caspase activation, Annexin V staining, etc.), and trafficking. Inclusion of such experiments are essential to verify the main conclusions of the study.
2. Overall, the effects of TNFR1 and/or TNFR2-deletion on NK cell abundance and maturation, and control of bacterial infection, are relatively modest.
3. The study would benefit from experiments that address a mechanistic explanation for the putative differences in TNFR1 vs. TNFR2 function in NK cells. For example, do the two receptors activate different downstream signaling pathways?
4. The study does not address whether TNFR1 and R2 are expressed by similar or different populations of NK cells in naïve and infected mice, and whether inflammatory signals regulate these expression patterns. Such information could provide insight into why the two receptors may exert unique effects on the bulk NK cell population.

Additional/minor concerns:

5. The term 'immature' NK cell (iNK) is broadly used in the field to designate a specific and early stage of mouse NK cell development that precedes the 'mature' NK cell (mNK) stage. iNKs are generally very rare in peripheral organs of naïve

mice and presumably nearly all of the NK cells analyzed in this manuscript are mNKs in varying stages of post-development maturation and differentiation. To avoid confusion, the authors should refer to CD27+CD11b- mNKs as 'less mature', rather than 'immature', throughout the manuscript.

6. The genes highlighted as identifiers of Cluster 4 in the Fig 1b, and of Cluster 5 in Fig. 4b, do not strongly support its designation as a "Proliferating" NK cell subset. Additional proliferation-associated genes should be shown to support this conclusion.

7. The text (Line 97) states that Tim3 was upregulated on NK cells in 'spleen and liver' of infected mice, but data are only shown for the spleen.

8. For flow cytometry-based assays, were ILC1s (e.g., Eomes- cells or CD49b- cells) excluded from the NK cell gates? On a related note, are ILC1s impacted by Tim3-deficiency/overexpression and TNFR1 and/or R2-deficiency?

9. Text (Line 109) states that there was a slight decrease in the percentage of M1 NK cells 'all organs' in the Tim3 strains, but data are only shown for the spleen

10. The interpretation of data in Fig. S3c-f that Tim3 deletion confers 'survival benefits' in NK cells is flawed. The data could be equally well explained by effects on proliferation and/or trafficking. The specific impact of Tim3-deficiency on NK cell survival, proliferation, and trafficking needs to be experimentally evaluated using assays specific to these different cellular processes.

11. The impact of Tim3 overexpression in NK cells on total bacterial loads is relatively modest (Fig. S4).

12. The statement (Line 121) that "we did not see increases in NK cell numbers" in the Ncr1-Cre x Tim3-fl/fl mice is misleading, since the data show a significant reduction in numbers in these mice.

13. For the studies in Figure 2A, it would be useful to evaluate the impact of IL-12 and IL-18 on Tim3 (and TNFR1/2) expression, since these cytokines play very important roles in NK cell activation.

14. The statement (Lines 153-154) that "there was no difference in NK cell numbers in the TNFR floxed mice (Fig. 3c)" is inconsistent with the statistically significant reduction ($p=0.014$) depicted in the figure.

15. The text (Lines 180) describes an "increased representation of neighborhoods within the proliferating clusters of WT NK cells compared to Tnfr2-/-, suggesting that TNFR2 may contribute to NK cell proliferation", but the corresponding figure(s) are not referenced and there is no information on the statistical tests used to support this conclusion (e.g., Fig. S6C and Figs. 4d-f lack statistics).

16. For studies in Fig 5, at least some of the DEGs could reflect differences in maturation status in WT vs. TNFR-deficient NK cells, rather than effects of TNFR signaling on specific genes per se. This issue could be addressed by performing RNAseq on sort-purified Immature, M1, and M2 populations from WT and TNFR-deficient NK cells.

17. Since only a few select genes are identified through annotation in Figs. 5a and S7b, it is difficult to determine the extent to which TNFR1- and TNFR2-deficiency have mutually exclusive effects on apoptosis and proliferation-related genes, respectively. This is a particular concern since the Pathway Analyses in Fig. 5b seem to suggest that both receptors impact proliferation and apoptosis-related pathways.

Version 2:

Reviewer comments:

Reviewer #1

(Remarks to the Author)

The authors have adequately addressed this reviewer's comments.

Reviewer #2

(Remarks to the Author)

The queries are answered and the experiments are performed by the Authors.

Reviewer #3

(Remarks to the Author)

The authors have satisfactorily addressed most comments from the first round of review. I have only a few minor critiques/suggestions for this revised manuscript:

1. Please indicate in the Methods section that the CellCycleScoring function in Seurat was used for cell cycle analyses on the scRNAseq data.
2. The text describing data from Fig. S4c-f (lines 126-129) should be revised to state that relative frequencies, not numbers, of Tim3-deficient and -overexpressing NK cells were analyzed in the adoptive transfer experiments. Since the data are depicted as relative frequencies, it is unclear whether Tim3-deficient NK cells are increasing or Tim3-overexpressing NK cells are decreasing in number (or both are occurring simultaneously), or both populations are increasing (or decreasing) in number but at different rates.
3. Please update legend for the blue histogram in Fig. S6g to reflect the "IL15+TNF (10ng/mL)" condition.
4. For data in Figure 4b, please describe how "Survival" was calculated in the Methods section.
5. For data in Fig. S8, the legend states that NK cells were "analyzed three days post onset of visible symptoms". Please clearly describe the specific visible symptoms that were used to define disease onset and clarify whether cells were collected from all mice 3 days after any single mouse displayed disease onset, or whether collection was relative to disease onset for each individual mouse.
6. For all text following line 231, "Tnfr1fl/fl" and "Tnfr2fl/fl" mice should be referred to as NcrcreTnfr1fl/fl and NcrcreTnfr2fl/fl, respectively (or Tnfr1-deficient and Tnfr2-deficient NK cells).
7. All text that states or implies that TNFR1 and/or TNFR2 act individually to promote NK cell proliferation should be carefully revised to remove this interpretation, since the functional assays in Fig. S15 (Edu incorporation and Ki67 protein expression) do not support this conclusion. Examples include statements in lines 289, 292, and 304.

McCulloch et al, Point-by-Point reply to Reviewer Comments

Dear Reviewers,

Thank you for your invaluable contributions to our submission. Your detailed comments and insightful feedback have been instrumental in enhancing the quality and depth of our work. We firmly believe that the study has significantly improved thanks to your thorough reviews. Thank you once again for your invaluable support and guidance.

Reviewer #1 (Remarks to the Author):

This manuscript presents a study investigating the role of tumor necrosis factor (TNF) in regulating natural killer (NK) cells during bacterial infection. The researchers identified Tim3 as a biomarker of TNF signaling in NK cells during infection. Mouse models were used to explore the functions of TNF receptors (TNFR1 and TNFR2) in NK cells during bacterial infection. Deletion of TNFR1 enhanced cell survival and immunity, while TNFR2 deletion limited NK cell function and worsened immunity. The study also observed similar patterns in NK cells from human sepsis.

However, there are several issues noted in the manuscript:

The effect of S. Typhimurium infection on NK cell differentiation, proliferation, and migration in the liver should be clarified. The elevated expression of Tim3 might be related to increased NK cell presence. Fig 1f needs clarification regarding the y-axis, whether it represents Tim3-positive NK cells in the total NK population or total spleen cells.

Thank you for your insightful comment regarding the effects of *S. Typhimurium* infection on NK cell dynamics in the liver. Addressing your query, we have previously published data demonstrating no significant difference in NK cell numbers in the liver of mice following *S. Typhimurium* infection (<https://onlinelibrary.wiley.com/doi/full/10.1111/imm.13516>). This finding leads us to believe that the infection does not notably influence NK cell proliferation in the liver. Concerning the expression of Tim3, it's important to note that this marker is not typically expressed in NK cells from naive mice in any organ, indicating that its presence is a response to specific stimuli rather than a baseline characteristic. In **Fig. 1f**, the y-axis specifically represents the percentage of NK cells expressing Tim3, not the total spleen cell population. This clarification is crucial as it indicates that the observed elevated expression of Tim3 is a response specific to NK cells and is independent of the total number of NK cells present. This observation suggests that Tim3 expression may be a significant marker of NK cell activation or differentiation in response to *S. Typhimurium* infection, rather than a mere consequence of altered NK cell quantities.

The impact of TNFR1 and TNFR2 deletion on the total NK cell population in the liver and spleen before and after infection needs clarification. Fig 3c shows a reduction in NK cell numbers, which contradicts the statement in the text that "there was no difference in NK cell numbers in the TNFR floxed mice."

Thank you for pointing out this inconsistency. Upon reviewing your comment, we realized that the version of **Fig. 3** included in our submission inadvertently displayed data from only one experimental repeat, leading to the observed contradiction. We have rectified this issue by updating **Fig. 3** to incorporate the results from a second, complementary experiment. The revised figure now accurately reflects the combined data, which clearly shows that there is no significant difference in NK cell numbers in the liver and spleen of TNFR1/TNFR2 floxed mice compared to WT controls. This aligns with the statement in our manuscript text.

The conclusion that TNF is important for NK cell maturation, differentiation, and proliferation should be supported by data obtained from non-infected mice as well, as all data were collected from mice infected with bacteria.

In order to address this important point, we included the immunophenotyping data from TNFR1 or TNFR2 floxed mice (**Figure S7**), whereby we observed a minor reduction in NK cell maturation after deletion of either receptor. However, obtaining data on proliferation in naïve mice is difficult, as NK cells need a stimulus to induce proliferation. *In vitro* proliferation assays are biased towards a TNFR1 response as addition of soluble TNF will robustly activate TNFR1 but poorly activate TNFR2 ([https://doi.org/10.1016/0092-8674\(95\)90192-2](https://doi.org/10.1016/0092-8674(95)90192-2)).

During the revisions, Prof. H. Wajant kindly provided to us the "TNFR2-specific agonist" NewSTAR2 published in *Frontiers in Immunology* (<https://pubmed.ncbi.nlm.nih.gov/35769484/>). This inhibitor was published as an agonist for TNFR2 using HT1080 cell lines transduced for high expression of TNFR2. In attempt to stimulate TNFR2 *in vitro* using this agonist in primary murine NK cells, these showed no difference in any readout measured (IFN- γ shown below in **Backup Data 1** as an example). Thus we are not confident that this agonist was functional.

Backup Data 1: TNFR2-agonist NewStar2 does not enhance IFN- γ production from NK cells. Total splenocytes from C57BL/6 mice were cultured for four hours in the presence of the indicated stimulations (see 'Splenocyte stimulation assay' in methods for more details). The percentage of NK cells expressing IFN- γ is shown.

The mechanism by which TNF induces Tim3 and its potential effects on Tim4 should be examined or discussed.

To the best of our knowledge, NK cells do not express Tim4. Indeed, reanalysis of our scRNA-seq dataset showed negligible levels of *Timd4* expression (gene encoding for Tim4, **Backup Data 2**). Indeed, we think this is an interesting thought and could be worth looking at in the context of macrophages, however this was outside of the scope of this publication.

Backup Data 2: Tim4 is not expressed in NK cell populations. NK cells from day 4 infected C57BL/6 spleens were analyzed by scRNA-seq. Expression of *Timd4* (encoding Tim4) across NK cell maturation clusters.

The authors suggest that TNFR1 or TNFR2 knockout down-regulates Tim3, and the underlying mechanisms for this regulation should be explored.

In response to this feedback, we have begun to further explore the mechanisms by which TNF induces function/dysfunction in NK cells by using specific inhibitors of TNF downstream pathways. Specifically, we employed small molecule inhibitors, BI605906 (IKK β inhibitor) and SP600125 (JNK inhibitor). As an example, in **Backup Data 3** we show that induction of IFN- γ production in NK cells by TNF is dependent on IKK β but not JNK. As we explored these experiments we realized that the intricacies of TNF signaling demand far more tools than we had access to for the purpose of this revisions. For example, we would require a functional TNFR2 agonist as well as further inhibitors of TNF regulated pathway such as B022 (NIK inhibitor), LY294002 (PI3K inhibitor), Losmapimod (p38 MAPK inhibitor), and potentially further mouse models of genetic deficiency to fully appreciate the underlying mechanisms. As an example, as soluble TNF

poorly activates TNFR2, **Backup Data 3** only tells us about how TNFR1 induces IFN- γ but cannot tell us about TNFR2. Further, it cannot tell us whether other TNF-induced pathways beyond IKK β and JNK synergize to induce IFN- γ . We propose to follow up on the mechanisms in collaboration with experts in TNF signalling in future studies, in which we can employ more tools to rigorously and comprehensively interrogate such mechanisms. We envisage this to be an entire project in itself that expands far beyond the scope of this study. However, these experiments would be critical for any translation of the findings presented here into clinical disease as they would allow us to appreciate and dissect the specific mechanisms at play to allow for specific targeting of pathways inducing dysfunction.

Backup Data 3: TNF induced production of IFN- is dependent on IKK β . Total splenocytes from C57BL/6 mice were cultured for four hours in the presence of IL-12 as well as TNF (10ng/ml), IKK inhibitor BI605906 (100 μ M), or JNK inhibitor SP600125 (100 μ M). The percentage of NK cells expressing IFN- γ is shown.

The differing effects of TNFR deletion observed in the spleen and liver, as shown in Fig 5e, should be discussed.

A discussion point has been added to lines 363-369, where we speculate this could be to differences in the architecture of TNF presentation in the spleen vs the liver.

The figure legend in Fig 5 does not match with the panel label, requiring correction.

Thank you for pointing out this inconsistency, this error has been corrected.

The authors propose that TNFR1 and TNFR2 regulate NK cell death/apoptosis and proliferation, respectively. It would be interesting to explore the expression pattern of TNFR1 and TNFR2 in different NK cell clusters using scRNA-seq data.

We looked at TNFR1 and TNFR2 expression across different NK cell clusters in our naïve dataset (which was not included in the paper). We saw very minor increases in TNFR1 and TNFR2 expression in the more mature NK cell subsets (**Backup Data 4a-b**). We also

looked at expression of TNFR1 and TNFR2 by flow cytometry across different maturation subsets, and found no differences (**Backup Data 4c-d**).

Backup Data 4: TNFR expression is unchanged in different NK cell populations. NK cells from naïve C57BL/6 spleens were analyzed by scRNA-seq. **a**, Expression of *Tnfr1sf1a* (encoding TNFR1) across NK cell maturation clusters. **b**, Expression of *Tnfr1sf1b* (encoding TNFR2) across NK cell maturation clusters. Splenic NK cells were subject to flow cytometry and TNFR1 or TNFR2 expression analyzed across maturation subsets. **c**, MFI of TNFR1 across less mature (CD11b-KLRG1-), M1 (CD11b+KLRG1-), or M2 (CD11b+KLRG1+) NK cells. **d**, Expression of TNFR2 across the same subsets.

We also looked at TFNR1 and TNFR2 gene expression in the infected datasets from **Figure 1a-d**, expression of both genes was fairly consistent across the different clusters, with very minor differences seen in the 3 proliferating clusters 1,2 and 4 (**Backup Data 5a,b**). Thus, we do not believe there is sufficient evidence to suggest there are substantial differences in expression patterns of either receptor across different NK cell subsets/clusters.

Backup Data 5: TNFR expression is similar across NK cell clusters in infected mice. Wild-type mice were infected with *S. Typhimurium* and NK cells analyzed by scRNA-seq on day four post-infection (seen in **Figure 1a-d** of the paper). **a**, Expression of *Tnfr1sf1a*

(encoding TNFR1) across NK cell clusters. **b**, Expression of *Tnfr1sf1b* (encoding TNFR2) across NK cell maturation clusters.

In Fig 6, it would be beneficial to compare the expression levels of TNFR1 and TNFR2 in NK cells of septic patients versus non-septic patients.

We agree that this is a useful analysis, however as shown in **Backup Data 6** expression of both TNFR1 and TNFR2 were unchanged across the different cohorts in the sepsis datasets.

Backup Data 6: TNFR expression is similar in NK cells across sepsis patients.

Reanalysis of scRNA-seq datasets of sepsis patients from Reyes *et al*, 2020, and seen in **Figure 6d-i** of this paper. **a**, Expression of *Tnfr1sf1a* (encoding TNFR1) in NK cells across the cohorts. **b**, Expression of *Tnfr1sf1b* (encoding TNFR2) in NK cells across the cohorts.

The distinct or opposite effects of TNFR1 and TNFR2 on other immune cells, such as macrophages, should be compared and discussed.

We have added a further discussion point to the opposing effects of TNFR1 and TNFR2 on other immune cells to lines 336-362

Minor issues, including typos and missing words, should be carefully reviewed and corrected.

We thank you for your support and valuable comments/suggestions. We have carefully tracked all changes made during the revision process.

Reviewer #2 (Remarks to the Author):

The Authors have elucidated the role of TNFR 1/2 in NK cells in the differential outcome of the bacterial infection in mice model.

The authors have interesting findings and will benefit from looking into the physiological relevance of the TNFR1/2- Tim3 - axis.

TNFR1/2 brings about differential outcomes where TNFR1 deletion promoted cell survival and proliferation of NK cells and TNF2R deletion limited cell proliferation

Thank you for your insightful suggestions and valuable feedback, which have significantly enriched our manuscript.

1. Percentage of NK cells expressing Tim3 is high post 2 days infection , which gradually falls. What is the status of TNFR1 and TNFR2 expression on the same subsets? What is the level of Serum TNF and IFN- γ at the same time period?

We have repeated the time-course to look at these parameters. TNFR1 expression in NK cells dramatically dropped at days 7-10 post infection, which could be due to specific death of TNFR1 expressing NK cells. This expression had recovered by day 14. Conversely, TNFR2 expression dramatically increased early in infection, before gradually returning to baseline levels. The increased TNFR2 expression is consistent with a very recent publication showing increased TNFR2 expression early after challenge with either LPS or MCMV (<https://www.nature.com/articles/s41423-023-01071-4>). This data has been included as **Fig S6**.

In the same experiment, we measured IFN- γ in both plasma and spleen and liver homogenates (**Backup data 7**). In the plasma, IFN- γ dramatically increased on day 7, returning to normal levels by day 21. In the spleens, it remained high between days 10 and 21, while in the livers, the levels increased on day 1 and plateaued at day 14.

Backup Data 7: IFN- γ titers across different organs. C57BL/6 mice were infected with *S. Typhimurium* and plasma, spleens, and livers taken at to quantify IFN- γ titers by ELISA. Spleens and livers were homogenized in 10ml and an aliquot of this organ homogenate take for ELISA: Levels of IFN- γ in each organ is shown.

2. *Survival Index of floxed mice (both TNFR1^{-/-} and 2^{-/-}, WT) must be scored to elucidate the final outcome of the infection process.*

It is important to note that the data presented in the first iteration of the paper uses an attenuated mutant of *Salmonella Typhimurium*, which is sublethal (<https://doi.org/10.1038/291238a0>). Thus, we do not expect survival differences in the different mouse lines in this model. To explore survival we would use the wild-type *S. Typhimurium* strain, SL1344. However, The University of Queensland (UQ) Animal Ethics Committee (AEC) has exceptionally strict policies on animal ethics, where survival experiments are unlikely to be justified and approved. Thus, we have not been able to get approval for survival experiments in this project. However, we believe that the differences in NK cell function and organ bacterial burdens adequately show the effects of the TNF receptors on NK cells in our model. While survival indexes would be interesting to examine the final outcome of infection, we agree with the UQ AEC that the harm this would cause to the animals is not justified.

3. *CFU analysis of bacterial burden in these floxed mice. Does the dissemination of Salmonella gets perturbed in either of the floxed mice?*

We show in **Figure 3g** that deletion of both TNFR1 and TNFR2 at the same time has no effect on bacterial burdens, however in **Figure 5g-h** we show that deletion of TNFR1 reduces bacterial burdens in the spleen, and deletion of TNFR2 increases bacterial burdens in the liver.

4. *In this particular article, Salmonella Typhimurium infection drives NK cell loss and conversion to ILC1-like cells, and CIS inhibition enhances antibacterial immunity, Timothy R. McCulloch, Gustavo R. Rossi, Timothy J. Wells, Fernando Souza-Fonseca Guimaraes doi:<https://doi.org/10.1101/2021.11.29.470332>, STM infection is reported to drive NK cell loss. One of the author Gustavo Rossi, is common in both the papers. It would be worthwhile to see that the loss of NK cells by STM is linked to TNFR1 and 2. In the floxed mice, the authors can look for the plasticity of NK cells.*

In **Figure 5e** we show that there are more NK cells in the spleen of TNFR1 KO mice, and less NK cells in the spleen of TNFR2 KO mice. We believe this data shows that this reviewer is correct, the NK cell loss we previously observed can be attributed (at least in part) to TNF, and in particular TNFR1.

We also tried looking at the plasticity of NK cells our mouse models. In our previous paper that the reviewer referenced, we found that at day 10 post infection there was considerable conversion of NK cells to 'ILC1-like'. In this paper, we focused on day 4 post infection as this was the peak of Tim3 expression in NK cells, and thus we predicted it to be the most interesting timepoint to investigate the role of TNF. At this timepoint, we did not observe any conversion to 'ILC1-like' in either the spleen or the liver, thus we also

did not observe any differences in conversion between the floxed models. We agree that this could be a fascinating area of study, however it was not the focus of this paper.

5. Physiological route of infection by oral gavaging must be performed to elucidate the role of NK cells. One experiment with the percentage of NK cells with Tim3, TNFR 1/2 will suffice.

We agree that it is important to recapitulate this phenotype using the natural route of *S. Typhimurium* infection. We performed this experiment by orally gavaging C57BL/6 mice with wild-type *S. Typhimurium* and analyzing three days post the onset of symptoms. Consistent with the invasive model, both Tim3 and TNFR2 were upregulated on NK cells in the spleen of infected mice. However, there was no change in expression of TNFR1. This data has been added to **Figure S1d-e** and **Figure S6c,d**.

6. Organ burden and Survival index must be demonstrated to convincingly show the importance of the dichotomy in the disease outcome.

The organ burden differences have been provided in **Figure 5g**, showing differences in organ burdens between WT, TNFR1-flox, and TNFR2-flox. Once again, we would like to reiterate that survival indexes could not be demonstrated due to ethical constraints.

7. Though the human data-set for sepsis is good to know, it does not add anything to this manuscript. For this data to be meaningful, the isolation PBMC must be pulsed with STM and the NK cells markers and Tim3 expression can be followed.

Coincidentally, the dataset described does exist from Bossel Ben-Moshe *et al.*, 2019 (<https://doi.org/10.1038/s41467-019-11257-y>). We were able to reanalyze this dataset to show that indeed Tim3 expression (*HAVCR2*) is increased on NK cells after the PBMC are pulsed with *S. Typhimurium*. This data was added as **Fig. S12**. However, we believe the sepsis dataset to still be relevant for this paper. While the cases of sepsis in the dataset were not caused by *Salmonella*, we believe they help explain the underlying biology of NK cells during inflammation, rather than the impact of *Salmonella* per se.

Reviewer #3 (Remarks to the Author):

This study by McCulloch et al uses mice with NK cell-specific deficiencies in Tim3, TNFR1, and TNFR2 to explore the roles of these factors in NK cell maturation, function, and host defense against bacterial infection in vivo. The data suggest that Tim3 expression is regulated downstream of TNFR signaling NK cells, but deletion of Tim3 in NK cells has little effect on NK cell development/maturation at baseline or control of S. Typhimurium infection in vivo. The major finding of the study is that while both TNFR1 and TNFR2 regulate NK cell abundance and maturation in vivo, the individual receptors may have distinct and somewhat opposing roles. Specifically, TNFR1 may act to limit NK cell survival and anti-bacterial activity, whereas TNFR2 may act to promote NK cell proliferation. The discovery of dichotomous roles for TNFR1 and TNFR2 in NK cell biology is intriguing and novel, and is expected to be of interest to the field. Overall, the manuscript is well-written and the data are clearly presented. Nevertheless, addressing the following concerns would significantly improve the depth of the study and strengthen the key conclusions of the manuscript.

We greatly appreciate the thoughtful suggestions and constructive feedback provided by the reviewer, which have been instrumental in enhancing the quality of our work. We acknowledge the reviewer's feedback and have made significant efforts in this revision. However, should the reviewer still find our efforts insufficient, we are prepared to conduct further experiments, subject to an extended deadline in the next year, to comprehensively address any potential study's gaps

Major criticisms:

1. The main finding that TNFR1 and TNFR2 have differential roles in regulating NK cell survival and proliferation, respectively, is suggested by the data but not rigorously verified. Indeed, the study generally lacks experiments and assays designed to specifically measure cellular proliferation (CTV dye dilution, BrDU incorporation assays), cell death (caspase activation, Annexin V staining, etc.), and trafficking. Inclusion of such experiments are essential to verify the main conclusions of the study.

We assessed the proliferation and apoptosis of NK cells in vitro using CTV dye dilution and Annexin V staining. Under normal culture conditions (just IL-15), the TNFR1-deficient NK cells exhibited similar proliferation and survival (as indicated by Annexin V and PI staining) compared to the WT cells (**Backup Data 8**). Addition of TNF to WT NK cells reduced the amount of proliferative cells (**Backup Data 8c**), likely due to increased cell death in this condition. Indeed, AnnexinV/PI staining showed increased apoptosis in WT cells treated with TNF (**Backup Data 8d**). Neither the reduction in proliferation nor increased apoptosis in response to TNF were apparent in TNFR1-deficient NK cells, indicating both were TNFR1 dependent (**Backup Data 8**). Together, this data supports our hypothesis that TNFR1 contributes to cell death with limited effect on proliferation.

Backup Data 8: TNF driven apoptosis is dependent on TNFR1. NK cells from wild-type (*Ncr1^{cre}*) or TNFR1-deficient (*Ncr1^{cre} x Tnfr1^{fl/fl}*) mice were stained with CTV then cultured in the presence of IL-15 (20ng/ml) and TNF (10ng/ml). After 72 hours, NK cells were stained for apoptosis with AnnexinV and PI staining. **a**, Representative histograms showing CTV dilution in WT cells. **b**, Representative histograms showing CTV dilution in TNFR1 deficient cells. **c**, Percentage of proliferated cells in each condition. **d**, Percentage of cells in late apoptosis (AnnexinV⁺PI⁺) in each condition is shown. Data are from one experiments (n = 5-6 biological replicates). Each dot represents one animal, error bars indicate mean ± SEM. Groups were compared using one-way ANOVA for where P>0.05 was deemed not significant.

Importantly, soluble TNF only poorly activates TNFR2. Therefore, to further address the roles of TNFR2 in vitro we employed the use of a previously published TNFR2 agonist, NewStar2, kindly provided by Prof. H. Wajant. Unfortunately, Newstar2 was not able to induce any phenotypic changes to NK cells in any assay performed (IFN- γ production included as an example in **Backup Data 1**), leading us to believe this agonist was not functional in our hands. Without a reliable TNFR2 agonist, we have not been able to specifically address the role TNFR2 plays on cell death or proliferation at this stage.

As *S. Typhimurium* is a systemic infection which is diffuse across multiple organs, obtaining data on trafficking in this model is difficult. However, work is underway by our group to explore the role of TNF in trafficking of NK cells into tumors (example in vitro data shown as **Backup Data 9** shows that TNF does not alter migration towards tumor

cells). We expect this to be thoroughly explored in a future cancer focused paper by our group.

Backup Data 9: Effects of TNF inhibition on in vitro migration of NK cells. MC38 cells were seeded at 1×10^5 cells/well in 24 well plates in DMEM media supplemented with 10% HI-FBS, 1% P/S, 1% GlutaMAX, 1% NEAA and 1% sodium pyruvate. One day later, DMEM media was replaced with 700 μ L of IMDM media supplemented with 10% HI-FBS, 1% P/S, 1% GlutaMAX, 0.1% 2-mercaptoethanol and 25 ng/mL human rIL-15. NK cells were isolated from spleens of C57BL/6 mice by negative selection using Miltenyi Biotec NK cell isolation mouse kit (130-115-818), and 1×10^5 NK cells were added in 200 μ L of IMDM media to the upper chamber (8 μ m pore size). After overnight incubation at 37 °C, the cells in the lower chamber were analysed by Flow cytometry. NK cells were defined as CD45+, NK1.1+, and were further classified as live (A) or dead (B).

2. Overall, the effects of TNFR1 and/or TNFR2-deletion on NK cell abundance and maturation, and control of bacterial infection, are relatively modest.

We concur with your assessment. While the observed effects are relatively modest in the specific context we explored, we believe that these findings are crucial in highlighting TNFR1 and TNFR2 as novel regulators of NK cell function. This insight opens up new avenues for understanding the subtleties of NK cell regulation and their role in immune responses. Furthermore, we anticipate that our study will serve as a catalyst for additional research into the impact of TNF on NK cells. We are particularly interested in exploring different pathologies and environmental contexts, where the role of TNF might be more pronounced or functionally diverse. Such investigations could reveal more substantial effects and broaden our understanding of NK cell biology in health and disease.

3. The study would benefit from experiments that address a mechanistic explanation for the putative differences in TNFR1 vs. TNFR2 function in NK cells. For example, do the two receptors activate different downstream signaling pathways?

We have attempted to address this comment with *in vitro* assays, however without a reliable TNFR2 agonist these experiments are flawed and lack the proper controls. As we

previously showed in response to reviewer #2, we tried to use an agonist for TNFR2 (NewStar2), but it did not induce a clear phenotypic change in NK cells in our experiments. However, based on our knowledge of TNFR signaling we believe much of the difference observed between TNFR1 and TNFR2 could be down to the death domain of TNFR1. This is further suggested by our finding that TNF induced apoptosis of NK cells in a TNFR1-dependent manner. We have added a discussion point in lines 342-347 which we hope addresses this point.

4. The study does not address whether TNFR1 and R2 are expressed by similar or different populations of NK cells in naïve and infected mice, and whether inflammatory signals regulate these expression patterns. Such information could provide insight into why the two receptors may exert unique effects on the bulk NK cell population.

As we previously showed in response to reviewer #1, TNFR1 and TNFR2 expression do not appear to be on different populations of NK cells.

As we previously showed in response to reviewer #2, TNFR1 expression in NK cells dramatically dropped at days 7-10 post infection, which could be due to specific death of TNFR1 expressing NK cells. This expression had recovered by day 14. Conversely, TNFR2 expression dramatically increased early in infection, before gradually returning to baseline levels. The increased TNFR2 expression is consistent with a very recent publication showing increased TNFR2 expression early after challenge with either LPS or MCMV (<https://www.nature.com/articles/s41423-023-01071-4>). These data were added as **Figure S6a,b**.

Additional/minor concerns:

5. The term 'immature' NK cell (iNK) is broadly used in the field to designate a specific and early stage of mouse NK cell development that precedes the 'mature' NK cell (mNK) stage. iNKs are generally very rare in peripheral organs of naïve mice and presumably nearly all of the NK cells analyzed in this manuscript are mNKs in varying stages of post-development maturation and differentiation. To avoid confusion, the authors should refer to CD27+CD11b- mNKs as 'less mature', rather than 'immature', throughout the manuscript.

We agree that this is an important distinction, and all NK cells in the periphery should be mature. We have made this change across all Figures and in the text.

6. The genes highlighted as identifiers of Cluster 4 in the Fig 1b, and of Cluster 5 in Fig. 4b, do not strongly support its designation as a "Proliferating" NK cell subset. Additional proliferation-associated genes should be shown to support this conclusion.

The genes highlighted in these clusters include *Ccnb2*, *Cenpa*, *Cenpf* which are all involved in cell division and thus suggestive of early proliferation. Importantly, typical proliferation genes such as *Mki67* were also identified as defining genes for both of

these clusters by the 'FindAllMarkers' function in Seurat. However, the way we have visualized these results in **Fig 1b** and **4b**, means that *Mki67* cannot be highlighted to define multiple clusters. We hope it is appreciated that there is significant overlap in the expression of genes between the proliferating clusters in **Fig 1b**, and the dot representing *Mki67* in Cluster 5 of **Figure 4b** shows a much higher expression compared to other clusters, both of which support the idea that these clusters are proliferative. We predict that the differences in the three proliferative clusters could be down to stages of proliferation (e.g. early vs late).

7. The text (Line 97) states that *Tim3* was upregulated on NK cells in 'spleen and liver' of infected mice, but data are only shown for the spleen.

The liver data has been added to **Figure S1c**.

8. For flow cytometry-based assays, were ILC1s (e.g., *Eomes*- cells or *CD49b*- cells) excluded from the NK cell gates? On a related note, are ILC1s impacted by *Tim3*-deficiency/overexpression and *TNFR1* and/or *R2*-deficiency?

For the scRNA-sequencing experiments, *CD49b* was not included in the panel for sorting. However, a small number of ILC1s were found but removed during analysis by expression of *Cd200r* and *Tnfsf10*. In all other flow-based assays, *CD49b* expression was used to exclude ILC1s. In the steady state, we found no impact on ILC1 numbers or phenotype in our genetic mouse models. During *Salmonella* infection, liver ILC1s are lost at our timepoint of analysis (day 4 post infection), and thus no differences were observed. This ILC1 loss may be due to inflammation-induced lactate, as previously observed in other models (<https://doi.org/10.1016/j.celrep.2020.107855>). We have previously observed conversion of conventional NK cells (*CD49b*⁺*CD49*⁻) to an ILC1-like phenotype (*CD49b*⁺*CD49*⁺) by day 10 post *Salmonella* infection (<https://doi.org/10.1111/imm.13516>). However, the experiments performed here were at day 4 post infection, no conversion was observed, and thus no differences were observed between mouse models.

9. Text (Line 109) states that there was a slight decrease in the percentage of M1 NK cells 'all organs' in the *Tim3* strains, but data are only shown for the spleen

As we looked across five different organs, we only showed the spleen as representative for clarity. We have added 'spleen shown as representative' where the Figure is referenced.

10. The interpretation of data in Fig. S3c-f that *Tim3* deletion confers 'survival benefits' in NK cells is flawed. The data could be equally well explained by effects on proliferation and/or trafficking. The specific impact of *Tim3*-deficiency on NK cell survival, proliferation, and trafficking needs to be experimentally evaluated using assays specific to these different cellular processes.

We agree with this, and the interpretation has been removed. As TNF was the main focus of this paper, we did not want to follow this Tim3 phenotype further, however we expect to include these experiments in a future study by our group.

11. The impact of Tim3 overexpression in NK cells on total bacterial loads is relatively modest (Fig. S4).

Thank you for your observation regarding the modest impact of Tim3 overexpression on NK cells as it relates to total bacterial loads, as depicted in **Fig. S4**. We concur with your assessment. This modest effect, combined with the fact that such high levels of Tim3 overexpression are unlikely to occur under natural physiological conditions, influenced our decision not to pursue this phenotype further in our study. We believe focusing on more physiologically relevant aspects will yield findings that are more applicable to natural immune responses.

12. The statement (Line 121) that “we did not see increases in NK cell numbers” in the Ncr1-Cre x Tim3-fl/fl mice is misleading, since the data show a significant reduction in numbers in these mice.

We agree, this statement was misleading. We have now clarified this observation in the next sentence (line 132)

13. For the studies in Figure 2A, it would be useful to evaluate the impact of IL-12 and IL-18 on Tim3 (and TNFR1/2) expression, since these cytokines play very important roles in NK cell activation.

We agree that the impact of IL-12 and IL-18 on Tim3 is of interest. We have data showing that both IL-12 and IL-18 are able to upregulate Tim3 expression. This data has been added as **Figure S5**.

A recent paper has explored the upregulation of TNFR1 and TNFR2 in NK cells in response to various cytokines. Notably, while TNFR1 was unchanged in response to any stimulation, both IL-12 and IL-18 were able to significantly upregulate TNFR2. These cytokines could be behind the increased TNFR2 expression we observed in the early stage of infection (**Figure S6b**).

14. The statement (Lines 153-154) that “there was no difference in NK cell numbers in the TNFR floxed mice (Fig. 3c)” is inconsistent with the statistically significant reduction ($p=0.014$) depicted in the figure.

This contradiction was due to an oversight where an old version of **Fig. 3** was included, which only displayed a single experimental repeat. This has been amended to include the second experiment. The combined data shows no difference in NK cell numbers in the TNFR1/TNFR2 flox mice compared to WT, as was indicated in the text.

15. The text (Lines 180) describes an “increased representation of neighborhoods within the proliferating clusters of WT NK cells compared to *Tnfr2*^{-/-}, suggesting that TNFR2 may contribute to NK cell proliferation”, but the corresponding figure(s) are not referenced and there is no information on the statistical tests used to support this conclusion (e.g., Fig. S6C and Figs. 4d-f lack statistics).

The figure is now appropriately referenced in the text. We also added the statistical test and FDR cut-off used for **Figure 4c-f** in the figure legend.

16. For studies in Fig 5, at least some of the DEGs could reflect differences in maturation status in WT vs. TNFR-deficient NK cells, rather than effects of TNFR signaling on specific genes per se. This issue could be addressed by performing RNAseq on sort-purified Immature, M1, and M2 populations from WT and TNFR-deficient NK cells.

Instead of performing additional sequencing experiments, we were able to perform differential expression on pseudobulk replicates in the scRNA-seq dataset. With this approach we were able to examine the differential gene expression between WT and TNFR-deficient NK cells in each cluster. Notably, the differential abundance of apoptosis/cell death associated genes in the TNFR1-flox NK cells was preserved across each cluster. Difference in proliferation genes in the TNFR2-flox NK cells was lost, which is likely due to differences in proportions of proliferating clusters rather than differences in the proliferating clusters themselves. These figures have been added as **Fig. S10** and **Fig. S11**.

17. Since only a few select genes are identified through annotation in Figs. 5a and S7b, it is difficult to determine the extent to which TNFR1- and TNFR2-deficiency have mutually exclusive effects on apoptosis and proliferation-related genes, respectively. This is a particular concern since the Pathway Analyses in Fig. 5b seem to suggest that both receptors impact proliferation and apoptosis-related pathways.

We have included the DEG tables used to generate the volcano plots in **Fig. 5a** as supplementary tables for transparency. We don't necessarily believe that the effects on TNFR1 and TNFR2 deficiency on apoptosis and proliferation are strictly mutually exclusive in NK cells. Rather we believe that each receptor likely makes contributions to apoptosis and proliferation but the overall response to TNFR1 is dominated by apoptosis, and response to TNFR2 dominated by proliferation. This is highlighted by the specific genes we highlighted in the volcano plots and culminates in the clear differences in overall numbers on splenic NK cells seen in **Fig. 5e**. By searching in the literature what is described to other cell types, also help explain these phenotypes. As examples, in macrophages, TNFR2 can contribute to cell death by sensitizing the cell to TNFR1 signalling (<https://doi.org/10.1038/cddis.2016.285>). In CD8⁺ T cells, TNFR2 contributes to activation induced cell death (AICD) which can be counteracted by TNFR1. Taking together, we believe that these evidences highlight a more intricate interplay

between TNFR1 and TNFR2 signalling in NK cells which we are excited to continuously study and elucidate in future work.

Response to reviewer comments

Manuscript No.: NCOMMS-23-28763A

Title: Dichotomous outcomes of TNFR1 and 2 signaling in NK cell-mediated immune responses during inflammation

Article Type: Article

Reviewer #1

Comment 1

This manuscript presents a study investigating the role of tumor necrosis factor (TNF) in regulating natural killer (NK) cells during bacterial infection. The researchers identified Tim3 as a biomarker of TNF signaling in NK cells during infection. Mouse models were used to explore the functions of TNF receptors (TNFR1 and TNFR2) in NK cells during bacterial infection. Deletion of TNFR1 enhanced cell survival and immunity, while TNFR2 deletion limited NK cell function and worsened immunity. The study also observed similar patterns in NK cells from human sepsis. However, there are several issues noted in the manuscript.

Author response: We thank the reviewer for taking the time to provide valuable feedback and for eloquently summarizing the study. We have improved the manuscript according to reviewer feedback (please see detailed responses to the comments below).

Comment 2

The effect of *S. Typhimurium* infection on NK cell differentiation, proliferation, and migration in the liver should be clarified. The elevated expression of Tim3 might be related to increased NK cell presence. Fig 1f needs clarification regarding the y-axis, whether it represents Tim3-positive NK cells in the total NK population or total spleen cells.

Author response: We thank the reviewer for their query regarding NK cell dynamics during infection, particularly to rule out increased Tim3 expression being related to increased NK cell presence. We have previously observed that *S. Typhimurium* infection induces differentiation of NK cells into ILC1-like cells in the spleen and liver (McCulloch *et al.*, 2022, PMID: 35611558). Here, we further show that *S. Typhimurium* induces both maturation (assessed by CD11b and KLRG1 expression) and proliferation (assessed by Ki67 expression) in both the spleen and liver. This reference regarding the differentiation and the new data showing maturation/proliferation has been added to the paper (data as **Figure S1b-g**). The following paragraph in red has been added to the text (lines 83-89) to give full context to the reader, and new data shown below.

*“We have previously shown that NK cells are significantly impaired during bacterial infection, including differentiation into ILC-like cells and organ-specific depletion¹⁷ Here, by infecting C57BL/6 mice with attenuated *S. Typhimurium* SL3261 (causing a chronic sublethal infection¹⁸) and using flow cytometry to phenotype the NK cells from spleen and liver (gating strategy shown in Fig. S1a), we find that bacterial infection also induces robust proliferation (Fig. S1b-c) and maturation (Fig. S1e-g) in both organs. However, the precise mechanisms underlying regulation of NK cells during bacterial infection are currently not well understood.”*

Fig. S1: Characterizing NK cells during *S. Typhimurium* infection. Wild-type C57BL/6 mice were infected with attenuated *S. Typhimurium* (SL3261) and NK cells analyzed in the spleens and livers at day four post infection. **a**, Representative gating strategy used throughout the paper to examine NK cells by flow cytometry. **b**, Representative flow cytometry plots showing spleen NK cell maturation, where “less mature” are defined as CD11b-KLRG1⁻, M1 defined as CD11b+KLRG1⁻, and M2 defined as CD11b+KLRG1⁺. **c,d**, Relative proportions of each NK cell maturation state in the spleen (**c**) or liver (**d**). **e**, Representative flow cytometry plots showing proliferation by Ki67 expression in spleen NK cells. **f,g**, Relative proportions of NK cells expressing Ki67 the spleen (**c**) or liver (**d**). Data in **b-g** from a single experiment ($n = 5$ biological replicates). Error bars indicate mean \pm SEM. Groups were compared using Mann-Whitney test, where $P < 0.05$ was considered statistically significant.

Concerning the expression of Tim3, it is important to note that this marker is not typically expressed in NK cells from naive mice in any organ, indicating that its increased expression during infection is truly in response to stimuli rather than change in NK cell migration/presence. This can be explained by a clarification of **Fig. 1f**, where the y-axis represents the percentage of NK cells expressing Tim3, not the total spleen cells, further confirming this observation was independent of increased NK cell presence. The y-axis label in **Fig. 1f** has been changed from “Spleen NK cells Tim3+ (%)” to “Spleen NK cells Tim3+ (% of total NK cells)” for clarity.

Comment 3

The impact of TNFR1 and TNFR2 deletion on the total NK cell population in the liver and spleen before and after infection needs clarification. Fig 3c shows a reduction in NK cell numbers, which contradicts the statement in the text that "there was no difference in NK cell numbers in the TNFR floxed mice."

Author response: We thank the reviewer for pointing out an inconsistency between the text and Figure 3. This issue was due to the initial submission inadvertently displaying an old version of Figure 3, which had a single experimental repeat, whereas the text was referencing the combined data of two complimentary experiments. This error has been rectified and the revised figure now accurately reflects the combined data, which clearly shows that there is no significant difference in NK cell numbers in the liver and spleen of TNFR1/TNFR2 floxed mice compared to WT controls. This aligns with the statement in our manuscript text.

Fig. 3: Suppression of TNF signaling in NK cells impacts their phenotype during infection. Transgenic *Ncr1cre* and *Ncr1creTnfr1fl/flTnfr2fl/fl* mice were infected with *S. Typhimurium* and spleens and livers analyzed on day 4 post-infection. **a**, Representative flow cytometry plots showing expression of Tim3 on splenic NK cells. **b**, Percentage of spleen or liver NK cells expressing Tim3. **c**, Total count of NK cells in the spleen of infected mice. **d**, Percentage of NK cells expressing Ki67. **e**, Percentage of splenic NK cells in each maturation stage based on CD11b and KLRG1 expression (Less mature = CD11b-KLRG1-, M1 = CD11b+KLRG1-, M2 = CD11b+KLRG1+). **f**, Percentage of NK cells expressing CD69. **g,h**, Bacterial burden per spleen (**g**) or liver (**h**), determined by CFU counts. Data are from two independent experiments ($n = 17$ biological replicates). Each dot represents one animal, error bars indicate mean \pm SEM. Groups were compared using Mann-Whitney test, where $P < 0.05$ was considered statistically significant.

Comment 4

The conclusion that TNF is important for NK cell maturation, differentiation, and proliferation should be supported by data obtained from non-infected mice as well, as all data were collected from mice infected with bacteria.

Author response: We thank the reviewer for this request for maturation, differentiation, and proliferation data from non-infected mice to complement the data from infected mice. In the initial version of the manuscript, we used “maturation” and “differentiation” interchangeably to describe what is commonly known as “maturation”. Upon consideration of your comment we considered our use of these terms inaccurate, and thus all references to TNF inducing NK cell differentiation have been removed. For clarity, and to address maturation, we have included the immunophenotyping data from naïve TNFR1 or TNFR2 floxed mice (**Figure S9**), where we observed a minor reduction in NK cell maturation after deletion of either receptor. We also performed *in vitro* experiments on NK cells isolated from naïve mice, where we treated them with IL-15 +/-TNF for 3 days and examined maturation by CD11b and KLRG1 staining. Notably, after three days in culture in just IL-15, the reduction in maturation in the floxed mice was preserved, further confirming the maturation deficit caused by TNFR1/TNFR2 deletion in NK cells. Treatment of WT NK cells with TNF led to a decrease in the M1 state (CD11b+KLRG1-) but increase in M2 (CD11b+KLRG1+), further suggesting that TNF contributes to NK cell terminal maturation. This effect was lost upon deletion of the TNFRs. This data has been added to the manuscript as Fig. S11 and the following text has been added to the manuscript in lines 245-249.

“Interestingly, these reductions in terminal maturation were also observed in naïve NK cells (Fig. S11a), suggesting that TNF acts on NK cells during development and the changes were not entirely due to infection-induced TNF. Culture of NK cells in the presence of TNF induced only minor increases in terminal maturation (Fig. S11b), which was lost upon deletion of TNFR signaling (Fig. S11c).”

Fig. S11: Maturation of TNFR-deficient NK cells. NK cells isolated from indicated genotypes were cultured in the presence of 25 ng/ml IL-15 +/- 10 ng/ml TNF for 3 days, then CD11b and KLRG1 expression measured to determine maturation status (less mature: CD11b-KLRG1-; M1: CD11b+KLRG1-; M2: CD11b+KLRG1+). a, Maturation status of each genotype treated with 25 ng/ml IL-15 only. b, Maturation status of NK cells from WT mice in response to IL-15 +/- TNF. c, Maturation status of NK cells from TNFR1/TNFR2-floxed mice in response to IL-15 +/- TNF. Data are from two independent experiments (n = 6 biological replicates). Each dot represents one animal, error bars indicate mean ± SEM. Groups were compared using one-

way ANOVA for a, or Wilcoxon matched-pairs rank test for b-c, where $P < 0.05$ was considered statistically significant.

To address proliferation from naïve mice, we reanalyzed the *in vitro* culture from **Fig 2a-b**, using CTV dye dilution to calculate the mean division number, as described by Hawkins *et al.*, 2007 (PMID: 17853861). We found addition of TNF at 5 or 10 ng/ml was able to promote the division of NK cells, indicating increased proliferation. Notably, this effect was not observed with 20 ng/ml, which we predict was due to the increased cell death in this condition. This data has been added as **Fig. S6g,h** (and the relevant panel shown below) and the following text added to the manuscript in lines 167-170.

“We also used CTV dye dilution to quantify NK cell proliferation and survival, as described previously²³. TNF at either 5 or 10 ng/ml was able to promote the division of NK cells (Fig. S6g,h), indicating enhanced proliferation. This effect was lost with the addition of TGF- β , which completely ablated proliferation (Fig. S6h).”

Fig. S6: TNF promotes NK cells proliferation in vitro. NK cells from wild-type C57BL/6 mice were isolated and cultured for three days in the presence of the 10 ng/ml IL-15 plus the indicated cytokines. g, Representative flow cytometry plots showing CTV dye dilution. h, Mean division number, calculated as described by Hawkins *et al.*²³. Data are from two independent experiments ($n = 6$ biological replicates). Each dot represents one animal, error bars indicate mean \pm SEM. Groups were compared using one-way ANOVA (comparisons to IL-15 only, * $P < 0.05$, ** $P < 0.01$, *** $P < 0.001$), where $P < 0.05$ was considered statistically significant.

Comment 5

The mechanism by which TNF induces Tim3 and its potential effects on Tim4 should be examined or discussed.

Author response: We thank the reviewer for this suggestion to investigate the mechanism of TNF induced Tim3 upregulation. To address this, we cultured NK cells with TNF in the presence of various inhibitors of the various TNF pathways. We inhibited used BI605906 (IKK- β inhibitor), SP600125 (JNK inhibitor), Losmapimod (p38 inhibitor), Amgen16 (NIK inhibitor), and rapamycin (mTOR inhibitor) to get a comprehensive overview of the mechanism of Tim3 upregulation. Notably, we found that Tim3 upregulation in response to TNF stimulation was critically dependent on IKK- β /NF- κ B but not JNK, p38, or NIK, suggesting that TNF induces Tim3 via the canonical NF- κ B pathway. These data have been added to **Fig. S6b-f** (and shown below) and the following explanation added to the text in lines 155-167.

“TNF acting via its receptors can activate multiple distinct signaling pathways, including canonical nuclear factor kappa-light-chain-enhancer of activated B cells (NF- κ B), non-canonical NF- κ B, and mitogen-activated protein kinases (MAPKs). To assess which of these pathways were required for Tim3 upregulation, NK cells were cultured with TNF in the presence of various TNF pathway inhibitors. Targeting canonical NF- κ B with a nuclear factor kappa-B kinase beta (IKK- β) inhibitor (BI60590624) was able prevent TNF-induced Tim3

upregulation in a dose dependent manner (Fig. S7b). Conversely, targeting MAPKs c-Jun N-terminal kinase (JNK, inhibited by SP60012525) and p38 (inhibited by Losmapimod26) showed no effect, nor did targeting non-canonical NF-kB with a NF-kB-inducing kinase (NIK) inhibitor (Amgen1627, Fig. S6c-e). mTOR inhibitor Rapamycin was also able to prevent Tim3 upregulation (Fig. S6f), likely due to the critical importance of mTOR in IL-15-mediated NK cell activation²⁸. Thus, these data show that TNF induces Tim3 upregulation via the canonical NF-kB pathway..”

Fig. S6: Upregulation of Tim3 in NK cells. NK cells from wild-type C57BL/6 mice were isolated and cultured for three days in the presence of the indicated cytokines and/or inhibitors. a, Expression of Tim3 on NK cells in response to IL-12 or IL-18 signaling. b, Effects of IKK-β inhibitor BI605906 on TNF-induced Tim3 expression. c, Effects of JNK inhibitor SP600125 on TNF-induced Tim3 expression. d, Effects of p38 inhibitor Losmapimod on TNF-induced Tim3 expression. e, Effects of NIK inhibitor Amgen16 on TNF-induced Tim3 expression. e, Effects of mTOR inhibitor rapamycin on TNF-induced Tim3 expression. Data in a are from one experiment (n = 4 biological replicates), and data from b-f are from two independent experiments (n = 6 biological replicates). Each dot represents one animal, error bars indicate mean ± SEM. Groups were compared using one-way ANOVA (*P<0.05, **P<0.01, ***P<0.001) where P < 0.05 was considered statistically significant.

The reviewer also asks about Tim4 expression. To the best of our knowledge, NK cells do not express Tim4. Indeed, reanalysis of our scRNA-seq datasets showed no expression of *Timd4* (gene encoding for Tim4, **Reviewer Data 1a**), whereas *Havcr2* (encoding Tim3) was expressed across multiple populations (**Reviewer Data 1b**). We further confirmed this with flow cytometry by staining NK cells from naive and infected spleens, using F4/80+ cells as a positive control (expecting most F4/80+ splenocytes to be macrophages, **Reviewer Data 1c**). Corroborating the scRNA-seq data, we found no expression of Tim4 on naive or infected NK cells, whereas 60-75% of spleen macrophages expressed Tim4 (**Reviewer Data 1d**). Indeed, we believe TNF-mediated regulation of Tim4 in macrophages could be worth investigating, however this is outside of the scope of this publication that focused on the NK cell compartment, and this new data has not been added to the manuscript.

Reviewer Data 1: Tim4 is not expressed in NK cell populations. NK cells from day 4 infected C57BL/6 spleens were analyzed by scRNA-seq. **a**, Expression of Timd4 (encoding Tim4) across NK cell clusters. **b**, Expression of Havcr2 (encoding Tim3) across NK cell clusters. NK cells and F4/80+ cells from day 4 infected C57BL/6 spleens were analyzed by flow cytometry. **c**, Representative gating strategy to examine NK cells and F4/80+ cells. **d**, Expression of Tim4 in indicated cell types. Data representative of 5 biological replicates.

Comment 6

The authors suggest that TNFR1 or TNFR2 knockout down-regulates Tim3, and the underlying mechanisms for this regulation should be explored.

We thank the reviewer for this suggestion, which we believe can be explained by the inhibitor data provided in **Fig. S6b-f**. The TNFRs are well characterized to activate NF- κ B, which we show in **Fig. S6b** to be critically important for the TNF-induced upregulation of Tim3. Deletion of either TNFR1 or TNFR2 would therefore result in decreased NF- κ B signaling, and subsequently reduced Tim3 expression.

Comment 7

The differing effects of TNFR deletion observed in the spleen and liver, as shown in Fig 5e, should be discussed.

Author response: We appreciate this suggestion to discuss the differing effects of TNFR deletion in the spleen/liver. A discussion point has been added to lines 420-431, where we speculate this could be to differences in the architecture of TNF presentation in the spleen vs the liver.

Lines 425-426: *“Interestingly, our results show that deletion of TNFR1 on NK cells improved bacterial burdens in the spleen, whereas deletion of TNFR2 worsened bacterial burdens in the liver.”*

Lines 432-436: *“Why these two receptors would be playing roles in different organs remains unclear, however could be due to differences in the presentation of TNF between the two organs. As TNFR1 can be activated by both soluble and membrane-bound TNF, whereas TNFR2 can only be activated by membrane-bound, our results may suggest an emphasis on soluble TNF in the spleen allowing the TNFR1 response to dominate.”*

Comment 8

The figure legend in Fig 5 does not match with the panel label, requiring correction.

Author response: We thank the reviewer for noticing the inconsistency between **Figure 5** and the legend, which has now been corrected.

Comment 9

The authors propose that TNFR1 and TNFR2 regulate NK cell death/apoptosis and proliferation, respectively. It would be interesting to explore the expression pattern of TNFR1 and TNFR2 in different NK cell clusters using scRNA-seq data.

Author response: This is a great suggestion by the reviewer to explore TNFR1 and TNFR2 expression in different NK cell clusters, and agree it could lead to interesting observations. Firstly, we looked at TNFR1 and TNFR2 expression across different NK cell clusters in the wild-type scRNA-seq from Figure 1, however differences across clusters were only minor. We also looked at expression of TNFR1 and TNFR2 by flow cytometry across different maturation subsets, however found no differences. This data has been added to the manuscript as **Fig. S7** (and shown below) and the following text added to lines 183-187

“Reanalysis of the scRNA-seq from Fig. 1 found that NK cells express both receptors for TNF, TNFR1 (encoded by Tnfrsf1a) and TNFR2 (encoded by Tnfrsf1b), although there did not appear to be differences in expression patterns across the clusters (Fig. S7a,b). This was confirmed in naïve mice by flow cytometry, where we found no difference in expression pattern of either receptor across the different maturation states (Fig. S7c,d).”

Fig. S7: TNFR expression across NK cell populations. Wild-type mice were infected with *S. Typhimurium* and NK cells analyzed by scRNA-seq on day four post-infection. a, Expression of *Tnfrsf1a*, encoding TNFR1, across NK cell clusters. b, Expression of *Tnfrsf1b*, encoding TNFR2, across NK cell clusters. Splenic NK cells from naïve mice were analyzed by flow cytometry. c, MFI of TNFR1 across NK cell populations. d, MFI of TNFR2 across NK cell populations (less mature: CD11b-KLRG1⁻; M1: CD11b+KLRG1⁻; CD11b+KLRG1⁺)

Comment 10

In Fig 6, it would be beneficial to compare the expression levels of TNFR1 and TNFR2 in NK cells of septic patients versus non-septic patients.

Author response: We thank the reviewer for what we once again agree to be a useful analysis. We performed this analysis, however expression levels of both TNFR1 and TNFR2 were unchanged across the different cohorts in the sepsis datasets (data added as **Fig. S16**). We also compared the percentage of NK cells expressing each gene across the patient groups and found some increases in % of NK cells expressing either *TNFRSF1A* or *TNFRSF1B* in septic patients compared to non-septic. This analysis has been added to **Fig. 6i,j**, and the following text added to the manuscript in lines 346-350.

“Examination of TNFRSF1A and TNFRSF1B, encoding TNFR1 and TNFR2 respectively, found that while the expression levels were unchanged (Fig. S16a,b) the percentage of NK cells expressing TNFR1 was increased in the URO group (Fig. 6i), whereas the percentage of NK cells expressing TNFR2 was increased in the Int-URO and URO groups (Fig. 6j), further supporting a potential role for TNF signaling.”

Figure S16: TNFR expression is similar in NK cells across sepsis patients. Reanalysis of scRNA-seq datasets of sepsis patients from Reyes et al, 2020. a, Expression of TNFRSF1B (encoding TNFR1) in NK cells across the cohorts. b, Expression of TNFRSF1B (encoding TNFR2) in NK cells across the cohorts.

Comment 11

The distinct or opposite effects of TNFR1 and TNFR2 on other immune cells, such as macrophages, should be compared and discussed.

Author response: We appreciate this suggestion and agree comparison with other cell types is important. We have added a further discussion point to compare the opposing effects of TNFR1 and TNFR2 on other immune cells to lines 410-424, as shown below.

“Individual roles for TNFR1 and TNFR2 have also been observed in other lymphocyte subsets. As an example, in CD4+ T cells, these two receptors oppose each other in the maintenance of autoimmunity. TNFR1 promotes the differentiation of inflammatory subsets of T cells, including Th1 and Th17, to exacerbate autoimmunity, whereas TNFR2 promotes the differentiation and function of protective Tregs. Curiously, CD8+ T cells appear different to our results in NK cells, whereby TNFR2 induces cell death through overactivation, and TNFR1 limits this response to promote survival. In contrast, in myeloid cells such as macrophages TNF acting through TNFR1 is a well characterized trigger of apoptosis and necroptosis. However, TNFR2 can also sensitize macrophages towards being receptive to necroptosis in response to TNFR1 signaling. In neutrophils, both TNFR’s promote neutrophil driven clearance of bacterial skin infection, where TNFR1 is responsible for trafficking of neutrophils to the site of infection but TNFR2 is responsible for direct antibacterial function. Taken together, clearly TNFR1 and TNFR2 play different roles in different immune cells, and more work is required to tease apart the relative contribution of each receptor in various cell types.”

Comment 12

Minor issues, including typos and missing words, should be carefully reviewed and corrected.

Author response: We are grateful to the reviewer for pointing out these mistakes. We have carefully reviewed the revised version of the manuscript to ensure that typos and missing words are corrected.

Reviewer #2

Comment 1

The Authors have elucidated the role of TNFR 1/2 in NK cells in the differential outcome of the bacterial infection in mice model. The authors have interesting findings and will benefit from looking into the physiological relevance of the TNFR1/2- Tim3 - axis. TNFR1/2 brings about differential outcomes where TNFR1 deletion promoted cell survival and proliferation of NK cells and TNFR2 deletion limited cell proliferation

Author response: We thank the reviewer for their encouraging feedback about the interesting findings. We have addressed the physiological relevance of the TNFR1/2- Tim3 – axis in the responses below.

Comment 2

Percentage of NK cells expressing Tim3 is high post 2 days infection, which gradually falls. What is the status of TNFR1 and TNFR2 expression on the same subsets? What is the level of Serum TNF and IFN- γ at the same time period?

Author response: We are grateful to the reviewer for their valuable comments regarding the expression of TNFR1 and TNFR2 on NK cells across the time-course of infection, as well as serum titers of TNF and IFN- γ . To address these matters, we repeated the time-course experiment and included TNFR1 and TNFR2 in the flow cytometry panel, as well as taking serum to quantify cytokine titers. TNFR1 expression in NK cells dramatically dropped at days 7-10 post infection, which could be due to specific death of TNFR1 expressing NK cells. This expression had recovered by day 14. Conversely, TNFR2 expression dramatically increased early in infection, before gradually returning to baseline levels. The increased TNFR2 expression is consistent with a very recent publication showing increased TNFR2 expression early after challenge with either LPS or MCMV (Khan *et al.*, 2024, PMID: 37553427). We also measured IFN- γ and TNF in serum. IFN- γ dramatically increased on day 7, returning to normal levels by day 21, whereas TNF peaked earlier at day 3 post-infection and stayed relatively elevated until returning to baseline at day 28. This data has been included as **Fig. S8a-d**, and the following added to the text in lines 187-196.

*“To assess the dynamics of TNFR expression on NK cells during *S. Typhimurium* infection, we examined the expression of TNFR1 and TNFR2 over the course of infection. While TNFR1 expression dropped at day 7-10 and recovered at day 14 (**Fig. S8a**), TNFR2 expression increased dramatically during early infection before gradually returning to baseline (**Fig. S8b**), showing inflammatory signals could modulate the expression of these receptors in NK cells. This data is consistent with a recent study showing LCMV infection and cytokines IL-12 and IL-18 could upregulate TNFR2 on NK cells. Cytokine analysis in the serum of these animals showed that TNF peaked early during infection (**Fig. S8c**), whereas IFN- γ peaked at day 7-10 before returning to baseline by day 21 (**Fig. S8d**).”*

Fig. S8: Expression of TNFR1 and TNFR2 on NK cells during *S. Typhimurium* infection. Wild-type mice were infected with attenuated *S. Typhimurium* (SL3261) and splenic NK cells analyzed by flow cytometry at different timepoints post-infection. a, Expression of TNFR1 on NK cells at different timepoints. b, Expression of TNFR2 on NK cells at different timepoints. c, Serum titers of IFN- γ at different timepoints. d, Serum titers of TNF at different timepoints. Wild-type C57BL/6 mice were infected with wild-type *S. Typhimurium* (SL1344) by oral gavage and NK cells analyzed three days post onset of visible symptoms. e-f, Expression of TNFR1 on splenic NK cells. g-h, Expression of TNFR2 on splenic NK cells. i-j, Expression of Tim3 on splenic NK cells. Data from a single experiment (n = 4-7 biological replicates) Error bars indicate mean \pm SEM. Groups were compared using Mann-Whitney test, where P < 0.05 was considered statistically significant.

Comment 3

Survival Index of floxed mice (both TNFR1^{-/-} and 2^{-/-}, WT) must be scored to elucidate the final outcome of the infection process.

Author response: We appreciate the reviewers comment regarding examining survival of the floxed mice. It is important to note that the data presented in the first iteration of the paper uses an attenuated mutant of *S. Typhimurium*, SL3261, which is sublethal in mice (Hoiseh and Stocker, 1981, PMID: 7015147). Thus, we do not expect survival differences in the different mouse lines in this model. To explore survival, we would use the wild-type *S. Typhimurium* strain SL1344; however, due our institutional animal ethics policy, which emphasize minimizing animal suffering, we have been unable to obtain approval for survival experiments in this project. Regardless, we believe that the differences in NK cell function and organ bacterial burdens adequately show the effects of the TNF receptors on NK cells in our model. We agree that survival experiments could add further depth to these studies, and these matters have been discussed in the revised version of the manuscript (lines 426-432), as shown below.

*“The strain of *S. Typhimurium* used in this study (SL3261) induces a chronic infection which is sublethal in mice, and so animal survival has not been scored. However, we believe the differences in NK cell function and bacterial burden we show strongly demonstrate the effects of the TNFRs on NK cells in our model. An interesting future experiment could be to explore*

the roles of the TNFRs on NK cells in an acute/lethal model, to determine how the differences in organ burden translate to animal survival.”

Comment 4

CFU analysis of bacterial burden in these floxed mice. Does the dissemination of Salmonella gets perturbed in either of the floxed mice?

Author response: We thank the reviewer for this question regarding whether the bacterial burdens is changed in the TNFR floxed mice. We show in **Figure 3g,h** that deletion of both TNFR1 and TNFR2 in NK cells at the same time has no effect on bacterial burdens, however in **Figure 5j-k** we show that deletion of TNFR1 reduces bacterial burdens in the spleen, and deletion of TNFR2 increases bacterial burdens in the liver.

Comment 5

In this particular article, Salmonella Typhimurium infection drives NK cell loss and conversion to ILC1-like cells, and CIS inhibition enhances antibacterial immunity, Timothy R. McCulloch, Gustavo R. Rossi, Timothy J. Wells, Fernando Souza-Fonseca Guimaraesdoi:<https://doi.org/10.1101/2021.11.29.470332>, STM infection is reported to drive NK cell loss. One of the author Gustavo Rossi, is common in both the papers. It would be worthwhile to see that the loss of NK cells by STM is linked to TNFR1 and 2. In the floxed mice, the authors can look for the plasticity of NK cells.

Author response: We appreciate the reviewers interest in our previous work, and this suggestion to explore the TNFRs in the context of our past findings. In **Fig. 5e** we show that there are more NK cells in the spleen of TNFR1 KO mice, and less NK cells in the spleen of TNFR2 KO mice. In **Fig. 5g-h** we show reduced apoptosis in TNFR1 deficient NK cells. Combined we believe this data shows that the NK cell loss we previously observed can be attributed (at least in part) to TNF, and in particular TNFR1. We have added the following text to lines 395-397 of the discussion to highlight this point.

“Regardless, we have previously observed an organ specific NK cell loss following S. Typhimurium infection, which the current study suggests can be attributed in part to TNFR1 signaling inducing cell death.”

We also tried looking at the NK-ILC plasticity in our mouse models. In our previous paper that the reviewer referenced, we found that at day 10 post infection there was considerable conversion of NK cells to ‘ILC1-like’. In this paper, we focused on day 4 post-infection as this was the peak of Tim3 expression in NK cells, and thus we predicted it to be the most interesting timepoint to investigate the role of TNF. At this timepoint, we did not observe any upregulation of CD49a, characteristic of conversion to ‘ILC1-like’, in either the spleen or the liver, thus we also did not observe any differences in conversion between the floxed models (liver shown as an example in **Reviewer Data 2**). Whether the TNFRs might contribute to NK cell plasticity is unknown and should be investigated further in future studies, particularly in the context of bacterial infection and cancer.

Reviewer Data 2: Group 1 ILC plasticity in TNFR flox mice. Group 1 ILC from naïve C57BL/6 livers were analyzed by flow cytometry. **a-c**, Representative flow cytometry plots showing CD49a and CD49b expression in group 1 ILC (CD45⁺Lineage⁻CD3⁺NKp46⁺NK1.1⁺) from WT (**a**), TNFR1-flox (**b**), or TNFR2-flox (**c**) mice.

Comment 6

Physiological route of infection by oral gavaging must be performed to elucidate the role of NK cells. One experiment with the percentage of NK cells with Tim3, TNFR 1/2 will suffice.

Author response: We thank the reviewer for this suggestion, and agree that it is important to recapitulate this phenotype using the natural route of *S. Typhimurium* infection. We performed this experiment by orally gavaging C57BL/6 mice with wild-type *S. Typhimurium* and analyzing four days after the onset of symptoms. Consistent with the invasive model, both Tim3 and TNFR2 were upregulated on NK cells in the spleen of infected mice. However, there was no change in expression of TNFR1. This data has been added to **Figure S8e-j** (relevant panels shown below) and the following explanation added to lines 196-199 of the manuscript.

“Additionally, we observed in an oral infection model that TNFR1 expression remained unchanged (Fig. S8e,f) while TNFR2 expression was increased in NK cells during infection (Fig. S8g,h). Tim3 was also upregulated on splenic NK cells following oral infection (Fig. S8i,j), further corroborating our findings..”

Fig. S8: Expression of TNFR1 and TNFR2 on NK cells during *S. Typhimurium* infection. Wild-type C57BL/6 mice were infected with wild-type *S. Typhimurium* (SL1344) by oral gavage and NK cells analyzed three days post onset of visible symptoms. e-f, Expression of TNFR1 on splenic NK cells. g-h, Expression of TNFR2 on splenic NK cells. i-j, Expression of Tim3 on splenic NK cells. Data from a single experiment ($n = 4-7$ biological replicates) Error bars indicate mean \pm SEM. Groups were compared using Mann-Whitney test, where $P < 0.05$ was considered statistically significant.

Comment 7

Organ burden and Survival index must be demonstrated to convincingly show the importance of the dichotomy in the disease outcome.

Author response: The organ burden differences have been provided in **Fig. 5j-k**, showing differences in organ burdens between WT, TNFR1-flox, and TNFR2-flox. Please see response to comment 3 for discussion on survival indexes.

Comment 8

Though the human data-set for sepsis is good to know, it does not add anything to this manuscript. For this data to be meaningful, the isolation PBMC must be pulsed with STM and the NK cells markers and Tim3 expression can be followed.

Author response: We appreciate the reviewer's thoughtful suggestion to examine PBMC pulsed with *S. Typhimurium*. To address this, we have reanalyzed a dataset from Bossel Ben-Moshe *et al.*, 2019 (PMID: 31332193). We were able to reanalyze this dataset to show that indeed Tim3 expression (*HAVCR2*) is increased on NK cells after the PBMC are pulsed with *S. Typhimurium*. This data was added to **Figure 6a-c**, and the following description added to lines 321-329 of the manuscript.

*“Thus far, this study has shown that TNF regulates NK cell function during bacterial infection in mice. We next investigated whether this could be the case in human disease by using a published scRNA-seq dataset from peripheral blood mononuclear cells (PBMCs) of human donors pulsed with *S. Typhimurium*31. We could identify a population of NK cells in this*

dataset (Fig. 6a), which had high expression of HAVCR2, encoding Tim3 (Fig. 6b). Comparison of naïve PBMC to those exposed to *S. Typhimurium* found that exposure to *S. Typhimurium* increased the expression of HAVCR2 on NK cells (Fig. 6c). These data show that *S. Typhimurium* can induce expression of Tim3 on human NK cells, which could potentially be through inducing TNF expression from other cell types.”

Fig. 6: scRNA-seq reveals Tim3 and TNF signaling in NK cells from human PBMC. Reanalysis of scRNA-seq from PBMC pulsed with *S. Typhimurium*. a, UMAP showing total cells from PBMC cultures. b, UMAP showing HAVCR2 expression in PBMC. c, Relative proportions of HAVCR2- or HAVCR2+ NK cells from naïve or *S. Typhimurium* pulsed PBMC.

Reviewer #3

Comment 1

This study by McCulloch *et al* uses mice with NK cell-specific deficiencies in Tim3, TNFR1, and TNFR2 to explore the roles of these factors in NK cell maturation, function, and host defense against bacterial infection *in vivo*. The data suggest that Tim3 expression is regulated downstream of TNFR signaling NK cells, but deletion of Tim3 in NK cells has little effect on NK cell development/maturation at baseline or control of *S. Typhimurium* infection *in vivo*. The major finding of the study is that while both TNFR1 and TNFR2 regulate NK cell abundance and maturation *in vivo*, the individual receptors may have distinct and somewhat opposing roles. Specifically, TNFR1 may act to limit NK cell survival and anti-bacterial activity, whereas TNFR2 may act to promote NK cell proliferation. The discovery of dichotomous roles for TNFR1 and TNFR2 in NK cell biology is intriguing and novel, and is expected to be of interest to the field. Overall, the manuscript is well-written and the data are clearly presented. Nevertheless, addressing the following concerns would significantly improve the depth of the study and strengthen the key conclusions of the manuscript.

Author response: We thank the reviewer for their encouraging feedback about the findings being intriguing and novel and of interest to the field. We are also happy to hear that the reviewer perceived the manuscript as well-written, and the data as clearly presented. We found the reviewers constructive comments valuable and we have addressed them below.

Major criticisms:

Comment 2.

The main finding that TNFR1 and TNFR2 have differential roles in regulating NK cell survival and proliferation, respectively, is suggested by the data but not rigorously verified. Indeed, the study generally lacks experiments and assays designed to specifically measure cellular proliferation (CTV dye dilution, BrDU incorporation assays), cell death (caspase activation, Annexin V staining, etc.), and trafficking. Inclusion of such experiments are essential to verify the main conclusions of the study.

Author response: This was a very valuable suggestion by the reviewer. To address this, we have assessed the cell death using Annexin V staining, which showed that TNFR1 deficient NK cells had less cell death compared to WT. This data has been added to **Figure 5g-h** and the relevant panel displayed below. We also assessed proliferation of NK cells by EdU incorporation and Ki67 staining. Curiously, our results from this staining did not replicate the observations of reduced *Mki67* expression in the scRNA-seq, where we saw no difference in either assay (data added to **Fig S15**). This could be because proliferation in our model reaches roughly 80% of NK cells, which is likely maxed out from other signals. Even upon deletion of both TNF receptors, the reduction in Ki67 staining is subtle (**Fig 3d**). We believe to fully tease out the different receptors we would need to develop an *in vitro* model of TNFR1 and TNFR2 stimulation. However, *in vitro* testing with TNF comes with an important caveat that soluble TNF poorly activates TNFR2. We have attempted to use a previously published TNFR2 agonist, NewStar2, kindly provided by Prof. H. Wajant (Vargas et al., 2022, PMID: 35769484). Unfortunately, Newstar2 was not able to induce any phenotypic changes to NK cells in any assay performed, leading us to believe this agonist was not functional in our hands. TNFR2 agonists are not commercially available and without one we have not been able to specifically address the role TNFR1 and TNFR2 play on proliferation at this stage. To account for this, we have slightly altered the paper to focus more on the dichotomous **outcomes** of TNFR1 and TNFR2 signaling (such as NK cell accumulation and bacterial burdens), which our data rigorously shows and hypothesizes acts through cell death and proliferation, respectively. The changes to the manuscript are detailed below.

Lines 305-308: “We used AnnexinV staining to further confirm that TNFR1, but not TNFR2, contributed to cell death during *S. Typhimurium* infection (Fig. 5g,h). Curiously, when examining proliferation by either EdU incorporation (Fig. S15a,b) or Ki67 staining (Fig. S15c,d) we found no differences between TNFR2-deficient and WT cells.”

Fig. 5: TNFR1 and TNFR2 have overlapping and opposing effects on NK cells. **g**, Representative flow cytometry plots showing AnnexinV and Viability staining. **h**, Percentages of apoptotic NK cells (defined as AnnexinV+Viability+ and displayed as relative to *Ncr1^{cre}* to normalize between experiments). Data from **g-h** are from two independent experiments (n = 11-12 biological replicates). Each dot represents one animal, error bars indicate mean \pm SEM. Groups were compared using Mann-Whitney *U*-test where $P < 0.05$ was considered statistically significant.

Title: “Dichotomous *outcomes* of TNFR1 and 2 signaling in NK cell-mediated immune responses during inflammation”

Abstract: “While both receptors contribute to NK cell maturation, interferon (IFN)- γ production, and upregulation of Tim3, the deletion of TNFR1 promoted cell survival and subsequently enhanced immunity to infection. Conversely, deletion of TNFR2 limited NK cell *accumulation* and effector function, worsening immunity.”

Discussion, lines 389-390: “A major finding from our study was the dichotomous outcomes of TNFR1 and TNFR2 in NK cells, where TNFR1 led to NK cell death while TNFR2 promoted accumulation.”

Discussion, lines 395-410: “Regardless, we have previously observed an organ specific NK cell loss following *S. Typhimurium* infection, which the current study suggests can be attributed in part to TNFR1 signaling inducing cell death. Due to the shared downstream signaling pathways of each receptor, it is also likely there is significant overlap between each receptor, whereby each contribute to cell death and proliferation to some degree. This is supported by our GSEA results which found deletion of TNFR1 or TNFR2 had impacts on genes associated with apoptosis and E2F signaling (predicting proliferation). However, we observed stark differences in overall NK cell numbers upon deletion of either receptor. This dichotomy is likely due to differences in the downstream signaling of TNFR1 and TNFR2. In particular, a primary difference is that TNFR1 contains a death domain which is able to trigger cell death via activation of caspases. These differences in signaling between TNFR1 and TNFR2 allowed each receptor to have a different outcome when acting on NK cells.”

We also agree that investigating trafficking is an important aspect, especially considering deletion of TNFR led to an upregulation of chemokine receptor *Cxcr3*. As *S. Typhimurium* is a systemic infection which is diffuse across multiple organs, obtaining data on trafficking in this model is difficult. Use of a photoconverting mouse model, such as the Kaede mice recently used to investigate tumor-infiltrating NK cells (Dean et al., 2024, PMID: 38267402), could address this issue, however we did not have access to this mouse model. The below discussion point in red was added to the discussion in lines 391-395 to highlight this. Work is underway by our group to explore the role of TNF in trafficking of NK cells into tumors (example

in vitro data shown as **Reviewer Data 4** shows that TNF does not alter migration towards tumor cells). We expect this to be thoroughly explored in a future cancer focused paper by our group.

Lines 391-395: *“Importantly, one aspect not investigated in this study is NK cell migration/trafficking, which could be playing an important role in the differences in NK cell accumulation in our various mouse models. Addressing this would require transgenic mouse models such as the photoconverting Kaede model43 or models of homing receptor deficiency, which we did not have access to for the purpose of this study.”*

Reviewer Data 4: Effects of TNF inhibition on in vitro migration of NK cells. MC38 cells were seeded at 1 x 10⁵ cells/well in 24 well plates in DMEM media supplemented with 10% HI-FBS, 1% P/S, 1% GlutaMAX, 1% NEAA and 1% sodium pyruvate. 24 hours later, DMEM media was replaced with IMDM media supplemented with 10% FBS, 1% P/S, 1% GlutaMAX, 0.1% 2-mercaptoethanol and 25 ng/ml human rIL-15 with or without murine TNF (10ng/ml) and SP600125 (JNK inhibitor, 10 μM). NK cells were isolated from spleens of C57BL/6 mice and 1 x 10⁵ NK cells were added in IMDM media to the upper chamber (8 μm pore size). After overnight incubation at 37 °C, the cells in the lower chamber were analyzed by flow cytometry. NK cells were defined as CD45+, NK1.1+, and were further classified as live (A) or dead (B). Data are from a single experiment (n = 4 biological replicates).

Comment 3

Overall, the effects of TNFR1 and/or TNFR2-deletion on NK cell abundance and maturation, and control of bacterial infection, are relatively modest.

Author response: We appreciate the reviewer’s observation that the observed effects are modest. While the effects may be relatively modest in the specific context we explored, we believe that these findings are crucial in highlighting TNFR1 and TNFR2 as novel regulators of NK cell function. This insight opens up new avenues for understanding the subtleties of NK cell regulation and their role in immune responses. We anticipate that our study will serve as a catalyst for additional research into the impact of TNF on NK cells, and the novel mouse models generated here will be exceptionally powerful in these studies. Thus, while our observed effects are modest, we still have great expectation that they will be of great interest to the field.

Comment 4

The study would benefit from experiments that address a mechanistic explanation for the putative differences in TNFR1 vs. TNFR2 function in NK cells. For example, do the two receptors activate different downstream signaling pathways?

Author response: We thank the reviewer for this suggestion. Based on our knowledge of TNFR signaling we believe much of the difference observed between TNFR1 and TNFR2 could be down to the death domain of TNFR1 (TRADD), which recruits RIP1, FADD and caspase-8 to induce apoptosis (Micheau and Tschopp, 2003, PMID: 12887920). This is further suggested by our finding that TNF induced apoptosis of NK cells in a TNFR1-dependent manner. We have added a discussion point in lines 400-405 to address this point, as copied below.

“However, we observed stark differences in overall NK cell numbers upon deletion of either receptor. This dichotomy is likely due to subtle differences in the downstream signaling of TNFR1 and TNFR2. In particular, a primary difference is that TNFR1 contains a death domain which is able to trigger cell death via activation of caspases⁴⁴. These differences in signaling between TNFR1 and TNFR2 allowed each receptor to have a different outcome when acting on NK cells.”

Comment 5

The study does not address whether TNFR1 and R2 are expressed by similar or different populations of NK cells in naïve and infected mice, and whether inflammatory signals regulate these expression patterns. Such information could provide insight into why the two receptors may exert unique effects on the bulk NK cell population.

Author response: This is a great suggestion by the reviewer to explore TNFR1 and TNFR2 expression in different NK cell populations. Firstly, we looked at TNFR1 and TNFR2 expression across different NK cell clusters in the wild-type scRNA-seq from Figure 1, however differences across clusters were only minor. We also looked at expression of TNFR1 and TNFR2 by flow cytometry across different maturation subsets, however found no differences. Thus, we believe there is no evidence to suggest the receptors are expressed on different populations on NK cells. This data has been added to the manuscript as **Fig. S7** and the following text added to lines 183-187.

*“Reanalysis of the scRNA-seq from Fig. 1 found that NK cells express both receptors for TNF, TNFR1 (encoded by *Tnfrsf1a*) and TNFR2 (encoded by *Tnfrsf1b*), although there did not appear to be differences in expression patterns across the clusters (**Fig. S7a,b**). This was confirmed in naïve mice by flow cytometry, where we found no difference in expression pattern of either receptor across the different maturation states (**Fig. S7c,d**).”*

Fig. S7: TNFR expression across NK cell populations. Wild-type mice were infected with *S. Typhimurium* and NK cells analyzed by scRNA-seq on day four post-infection. *a*, Expression of *Tnfrsf1a*, encoding TNFR1, across NK cell clusters. *b*, Expression of *Tnfrsf1b*, encoding TNFR2, across NK cell clusters. Splenic NK cells from naïve mice were analyzed by flow cytometry. *c*, MFI of TNFR1 across NK cell populations. *d*, MFI of TNFR2 across NK cell populations (less mature: CD11b-KLRG1⁻; M1: CD11b+KLRG1⁻; CD11b+KLRG1⁺)

To address whether inflammatory signals could regulate TNFR1 or TNFR2 expression, we examined TNFR1 and TNFR2 expression in NK cells across the course of infection. TNFR1 expression dramatically dropped at days 7-10 post infection, which could be due to specific death of TNFR1 expressing NK cells. This expression had recovered by day 14. Conversely, TNFR2 expression dramatically increased early in infection, before gradually returning to baseline levels. This data has been added to the manuscript as **Fig. S8a,b**. While we cannot be sure the exact inflammatory signals that are increasing TNFR2 expression in this context, the increased TNFR2 expression is consistent with a very recent publication showing increased TNFR2 expression early after challenge with either LPS or MCMV, or in vitro by stimulating with either IL-12 or IL-18. The following explanation has been added to the manuscript in lines 187-194.

*“To assess the dynamics of TNFR expression on NK cells during *S. Typhimurium* infection, we examined the expression of TNFR1 and TNFR2 over the course of infection. While TNFR1 expression dropped at day 7-10 and recovered at day 14 (Fig. S8a), TNFR2 expression increased dramatically during early infection before gradually returning to baseline (Fig. S8b), showing inflammatory signals could modulate the expression of these receptors in NK cells. This data is consistent with a recent study showing LCMV infection and cytokines IL-12 and IL-18 could upregulate TNFR2 on NK cells²⁹.*

Fig. S8: Expression of TNFR1 and TNFR2 on NK cells during *S. Typhimurium* infection. Wild-type mice were infected with attenuated *S. Typhimurium* (SL3261) and splenic NK cells analyzed by flow cytometry at different timepoints post-infection. a, Expression of TNFR1 on NK cells at different timepoints. b, Expression of TNFR2 on NK cells at different timepoints. c, Data from a single experiment (n = 4-7 biological replicates) Error bars indicate mean \pm SEM.

Additional/minor concerns:

Comment 6

The term 'immature' NK cell (iNK) is broadly used in the field to designate a specific and early stage of mouse NK cell development that precedes the 'mature' NK cell (mNK) stage. iNKs are generally very rare in peripheral organs of naïve mice and presumably nearly all of the NK cells analyzed in this manuscript are mNKs in varying stages of post-development maturation and differentiation. To avoid confusion, the authors should refer to CD27+CD11b- mNKs as 'less mature', rather than 'immature', throughout the manuscript.

Author response: We agree that this is an important distinction, and all NK cells in the periphery should be mature. We have made this change across all figures and in the text.

Comment 7

The genes highlighted as identifiers of Cluster 4 in the Fig 1b, and of Cluster 5 in Fig. 4b, do not strongly support its designation as a "Proliferating" NK cell subset. Additional proliferation-associated genes should be shown to support this conclusion.

Author response: We appreciate the reviewers comment regarding the proliferative clusters in each scRNA-seq dataset. The genes highlighted in these clusters include *Ccnb2*, *Cenpa*, *Cenpf* which are all involved in cell division and thus suggestive of early proliferation. Importantly, typical proliferation genes such as *Mki67* were also identified as defining genes for both of these clusters by the 'FindAllMarkers' function in Seurat. However, the way we have visualized these results in **Fig 1b** and **4b**, means that *Mki67* cannot highlighted to define multiple clusters. We hope it is appreciated that there is significant overlap in the expression of genes between the proliferating clusters in **Fig 1b**, and the dot representing *Mki67* in Cluster 5 of **Fig. 4b** shows a much higher expression compared to other clusters, both of which support the idea that these clusters are proliferative. Cell cycle analysis using the "CellCycleScoring" function in Seurat predicted that cluster 3 was in the S phase, whereas cluster 5 was in the G2M phase, giving further evidence these clusters were proliferative. We believe the cell cycle analysis to add further evidence that the queried NK cell clusters are proliferative. The following lines were added to the manuscript and cell cycle graphs added to **Fig S2c** and **Fig. S10b** (also shown below as **Reviewer Data 4**)

Lines 98-100: *"Cell cycle scoring predicted that the different proliferative clusters were in different stages of the cell cycle, where cluster 2 was mainly in S phase, and clusters 1 and 4 predominately G2/M phase (Fig. S2c)."*

Lines 228-230: *"As with the scRNA-seq from Fig. 1, the different proliferative clusters were predicted to be in different stages of the cell cycle, where cluster 3 was mostly assigned to S phase and clusters 2 and 5 predominantly assigned to G2/M phase (Fig. S10b)."*

Reviewer Data 4: Cell cycle predictions of the scRNA-seq. **a**, Cell cycle prediction of WT scRNA-seq using the CellCycleScoring function in Seurat. **b**, Cell cycle prediction of transgenic scRNA-seq using the CellCycleScoring function in Seurat.

Comment 8

The text (Line 97) states that Tim3 was upregulated on NK cells in ‘spleen and liver’ of infected mice, but data are only shown for the spleen.

Author response: We thank the reviewer for this observation. The liver data has now been added to **Figure S2d-e**, and the relevant panels shown below.

Fig. S2: Tim3 is upregulated on NK cells during S. Typhimurium infection. Wild-type C57BL/6 mice were infected with attenuated *S. Typhimurium* (SL3261) and NK analyzed at day four post infection. **d**, Representative flow plots showing expression of Tim3 on liver NK cells. **e**, Percentage of NK cells expressing Tim3 in the liver of naïve or *S. Typhimurium* infected mice. Data in **a-c** from a single experiment ($n = 4$ biological replicates) and **d,e** pooled from two independent experiments ($n = 14-15$ biological replicates). Error bars indicate mean \pm SEM. Groups were compared using Mann-Whitney test, where $P < 0.05$ was considered statistically significant.

Comment 9

For flow cytometry-based assays, were ILC1s (e.g., Eomes+ cells or CD49b+ cells) excluded from the NK cell gates? On a related note, are ILC1s impacted by Tim3-deficiency/overexpression and TNFR1 and/or R2-deficiency?

Author response: The reviewer raises an important point regarding the exclusion of ILC1 from experiments. For the scRNA-sequencing experiments, CD49b was not included in the panel for sorting. However, a small number of ILC1s were found but removed during analysis by

expression of *Cd200r* and *Tnfsf10*, and lack of *Eomes*. In all other flow-based assays, CD49b expression was used to exclude ILC1s. A representative gating strategy used in all *in vivo* experiments has been added to **Figure S1**, and the relevant panels shown below.

Fig. S1: Characterizing NK cells during *S. Typhimurium* infection. Wild-type C57BL/6 mice were infected with attenuated *S. Typhimurium* (SL3261) and NK cells analyzed in the spleens and livers at day four post infection. **a**, Representative gating strategy used throughout the paper to examine NK cells by flow cytometry.

The reviewer also raises an interesting point on the effect of Tim3 or TNFR transgenes on ILC1s. In the steady state, we found no impact on ILC1 numbers in the genetic mouse models (**Reviewer Data 5**). During *Salmonella* infection, liver ILC1s are lost at our timepoint of analysis (day 4 post infection, **Reviewer Data 5c-d**). Deletion of either TNFR1 or TNFR2 could not prevent this ILC1 loss, indicating it is not TNF dependent (**Reviewer Data 5d-f**). However ILC1 have previously been shown to be particularly sensitive to inflammation-induced lactate (<https://doi.org/10.1016/j.celrep.2020.107855>), which may be causing their loss in our model.

Reviewer Data 5: Liver ILC1s in the novel mouse models. Total ILC1 (CD45⁺Lineage⁻CD3⁻NKp46⁺CD49b⁻CD49a⁺) were counted in the livers of **a**, Tim3 genetic mouse models or **b**, TNFR genetic mouse models. Group 1 ILC (CD45⁺Lineage⁻CD3⁻NKp46⁺NK1.1⁺) from naïve or infected livers were analyzed by flow cytometry. Representative flow plots showing liver NK (Eomes⁺CD49b⁺) and ILC1 (Eomes⁻CD49b⁻) from naïve WT mice (**c**), infected WT mice (**d**), infected TNFR1-flox mice (**e**), and infected TNFR2-flox mice (**f**).

Comment 10

Text (Line 109) states that there was a slight decrease in the percentage of M1 NK cells ‘all organs’ in the Tim3 strains, but data are only shown for the spleen

Author response: We thank the reviewer for noticing this inconsistency between the text and figure. As we looked across five different organs, we only showed the spleen and liver as representative for simplicity. We have added ‘**spleen and liver shown as representative**’ where the Figure is referenced in line 121.

Comment 11

The interpretation of data in Fig. S3c-f that Tim3 deletion confers ‘survival benefits’ in NK cells is flawed. The data could be equally well explained by effects on proliferation and/or trafficking. The specific impact of Tim3-deficiency on NK cell survival, proliferation, and trafficking needs to be experimentally evaluated using assays specific to these different cellular processes.

Author response: We thank the reviewer for noticing this inaccuracy in our interpretation, which has been removed. Considering we found that NK-specific Tim3 deficiency did not improve bacterial clearance, this phenotype was not investigated further in our particular model. Additional work is currently underway in our group to investigate other pathologies in which Tim3 expression on NK cells can be important and have more relevance, e.g. cancer models. However, cancer models are beyond the scope of the current study and we plan to submit subsequent papers to explore these areas and incorporate the suggested experiments in future studies.

Comment 12

The impact of Tim3 overexpression in NK cells on total bacterial loads is relatively modest (Fig. S4).

Author response: We thank the reviewer for their observation regarding the modest impact of Tim3 overexpression on NK cells as it relates to total bacterial loads, as depicted in **Fig. S4**. This modest effect, combined with the fact that such high levels of Tim3 overexpression are unlikely to occur under natural physiological conditions, influenced our decision not to pursue this phenotype further in our study. We believe focusing on more physiologically relevant aspects will yield findings that are more applicable to natural immune responses.

Comment 13

The statement (Line 121) that “we did not see increases in NK cell numbers” in the Ncr1-Cre x Tim3-fl/fl mice is misleading, since the data show a significant reduction in numbers in these mice.

Author response: The reviewer makes an important observation, and we agree that this statement was inaccurate. We have now clarified this observation in the next sentence, with the changes shown below.

Line 133-138: “While we found that *Ncr1creFSF-Tim3* had reduced NK cell numbers (Fig. S5c), NK cell activation (by CD69 staining, Fig. S5d), and increased bacterial burdens (Fig. S5e), we did not observe the expected increases in NK cell numbers (Fig. S5c), changes in bacterial burdens (Fig. S5c), or differences in IFN-g in *Ncr1creTim3fl/fl* mice compared to controls. In fact, NK cells were surprisingly reduced in the livers of *Ncr1creTim3fl/fl* mice (Fig. S5c).

Comment 14

For the studies in Figure 2A, it would be useful to evaluate the impact of IL-12 and IL-18 on Tim3 (and TNFR1/2) expression, since these cytokines play very important roles in NK cell activation.

Author response: We agree that the impact of IL-12 and IL-18 on Tim3 is of interest. We have data showing that both IL-12 and IL-18 are able to upregulate Tim3 expression. This data has been added as **Figure S6**.

We have also shown that A recent paper has explored the upregulation of TNFR1 and TNFR2 in NK cells in response to various cytokines. Notably, while TNFR1 was unchanged in response to any stimulation, both IL-12 and IL-18 were able to significantly upregulate TNFR2. These cytokines could be behind the increased TNFR2 expression we observed in the early stage of infection (**Figure S8b**). This data has been added to the paper as Fig. S6 and the relevant panel shown below.

Lines 154-155: “We also observed that NK cell-related activating cytokines IL-12 and IL-18 were able to upregulate Tim3 (Fig. S6a).”

Fig. S6: Upregulation of Tim3 in NK cells. NK cells from wild-type C57BL/6 mice were isolated and cultured for three days in the presence of the indicated cytokines and/or inhibitors. a, Expression of Tim3 on NK cells in response to IL-12 or IL-18 signaling. Data in a are from one experiment (n = 4 biological replicates). Each dot represents one animal, error bars indicate mean \pm SEM. Groups were compared using one-way ANOVA (*P<0.05, **P<0.01, ***P<0.001) where P < 0.05 was considered statistically significant.

Comment 15

The statement (Lines 153-154) that “there was no difference in NK cell numbers in the TNFR floxed mice (Fig. 3c)” is inconsistent with the statistically significant reduction (p=0.014) depicted in the figure.

Author response: We thank the reviewer for pointing out an inconsistency between the text and Figure 3. This issue was due to the initial submission inadvertently displaying an old version of Figure 3, which had a single experimental repeat, whereas the text was referencing the combined data of two complimentary experiments. This has been rectified and the revised

figure now accurately reflects the combined data, which clearly shows that there is no significant difference in NK cell numbers in the liver and spleen of TNFR1/TNFR2 floxed mice compared to WT controls. This aligns with the statement in our manuscript text.

Fig. 3: Suppression of TNF signaling in NK cells impacts their phenotype during infection. Transgenic *Ncr1cre* and *Ncr1creTnfr1fl/flTnfr2fl/fl* mice were infected with *S. Typhimurium* and spleens and livers analyzed on day 4 post-infection. a, Representative flow cytometry plots showing expression of Tim3 on splenic NK cells. b, Percentage of spleen or liver NK cells expressing Tim3. c, Total count of NK cells in the spleen of infected mice. d, Percentage of NK cells expressing Ki67. e, Percentage of splenic NK cells in each maturation stage based on CD11b and KLRG1 expression (Less mature = CD11b-KLRG1-, M1 = CD11b+KLRG1-, M2 = CD11b+KLRG1+). f, Percentage of NK cells expressing CD69. g,h, Bacterial burden per spleen (g) or liver (h), determined by CFU counts. Data are from two independent experiments ($n = 17$ biological replicates). Each dot represents one animal, error bars indicate mean \pm SEM. Groups were compared using Mann-Whitney test, where $P < 0.05$ was considered statistically significant.

Comment 16

The text (Lines 180) describes an “increased representation of neighborhoods within the proliferating clusters of WT NK cells compared to *Tnfr2*^{-/-}, suggesting that TNFR2 may contribute to NK cell proliferation”, but the corresponding figure(s) are not referenced and there is no information on the statistical tests used to support this conclusion (e.g., Fig. S6C and Figs. 4d-f lack statistics).

Author response: We thank the reviewer for this observation, the figure is now appropriately referenced in the text. We also added the statistical test and false discovery rate (FDR) cut-off used for **Fig. 4c-f** in the figure legend.

Comment 17

For studies in Fig 5, at least some of the DEGs could reflect differences in maturation status in WT vs. TNFR-deficient NK cells, rather than effects of TNFR signaling on specific genes per se. This issue could be addressed by performing RNAseq on sort-purified Immature, M1, and M2 populations from WT and TNFR-deficient NK cells.

Author response: The reviewer has raised a good point. Instead of performing additional sequencing experiments, we were able to perform differential expression on pseudobulk replicates in the scRNA-seq dataset. With this approach we were able to examine the differential gene expression between WT and TNFR-deficient NK cells in each cluster. Notably, the differential abundance of apoptosis/cell death associated genes in the TNFR1-flox NK cells was preserved across each cluster. Difference in proliferation genes in the TNFR2-flox NK cells was largely lost, which is likely due to differences in proportions of proliferating clusters rather than differences in the proliferating clusters themselves. These figures have been added as **Fig. S13** and **Fig. S14**, and the following added to the text in lines 278-284.

*“Differential gene expression of each individual cluster found similar DEGs associated with apoptosis/cell death when comparing WT to *Tnfr1fl/fl*, reinforcing that DEGs in the bulk population were due to effects of TNFR1 signaling rather than changes in cluster proportions (Fig. S13-14). Conversely differences in proliferation genes between WT and *Tnfr2fl/fl* were largely lost when comparing different clusters, likely due to these differences being represented by changes in proportions of proliferating clusters (Fig. S13-14).”*

Comment 18

Since only a few select genes are identified through annotation in Figs. 5a and S7b, it is difficult to determine the extent to which TNFR1- and TNFR2-deficiency have mutually exclusive effects on apoptosis and proliferation-related genes, respectively. This is a particular concern since the Pathway Analyses in Fig. 5b seem to suggest that both receptors impact proliferation and apoptosis-related pathways.

Author response: The reviewer raises another important point. Firstly, we have included the DEG tables used to generate the volcano plots in **Fig. 5a** as supplementary tables for transparency. As discussed in Comment 2, the new data generated during this revision have allowed us to reframe the the roles of TNFR1 and TNFR2 on NK cells. In Fig S6 we show that recombinant TNF can induce NK cell proliferation *in vitro*, which is likely via TNFR1, as TNFR2 activation requires membrane bound TNF. Conversely, the scRNA-seq is highly suggestive that TNFR2 is the main driver of proliferation. Together these data suggest that both receptors play a role in proliferation. While the signaling of TNFR1 and TNFR2 likely overlaps, TNFR1 and TNFR2 signaling leads to dichotomous outcomes of NK cell accumulation, as demonstrated by the NK cell abundance data. We believe that each receptor likely makes contributions to apoptosis and proliferation but the overall response to TNFR1 is dominated by apoptosis, and response to TNFR2 dominated by proliferation. This is highlighted by the specific genes we highlighted in the volcano plots and culminates in the clear differences in overall numbers on splenic NK cells seen in **Fig. 5e**. Thus, TNFR1 and TNFR2 may not have mutually exclusive effects, but the outcomes of each receptor signaling of NK cell accumulation are clear. Description in the literature of the roles of each receptor on other cell types may also help explain these phenotypes. As examples, in macrophages, TNFR2 can contribute to cell death by sensitizing the cell to TNFR1 signaling. In CD8⁺ T cells, TNFR2

contributes to activation induced cell death (AICD) which can be counteracted by TNFR1. Taking together, we believe that this these evidence highlight a more intricate interplay between TNFR1 and TNFR2 signaling in NK cells which we are excited to continuously study and elucidate in future work. On top of the changes made in response to Comment 2, the following changes in red have been made in the manuscript to support these new conclusions.

Results lines 285-293: *“To investigate these mechanisms further we performed gene set enrichment analysis (GSEA) of relevant pathways. We found a significant reduction in the hallmark pathways for Mtorc1 signaling, apoptosis, and E2F targets in both knockouts compared to wild-type (Fig. 5b). Tnfr1fl/fl NK cells also had significantly reduced apoptosis signaling compared to Tnfr2 fl/fl cells, further suggesting that apoptosis was a dominant pathway triggered by TNFR1. That E2F targets was reduced in both knockouts suggests that both TNFRs could promote proliferation. Together these data are suggestive that TNFR1 and TNFR2 could play opposing roles, whereby both receptors promote proliferation and effector function in NK cells but TNFR1 simultaneously promotes cell death.”*

Discussion lines 405-419: *“Individual roles for TNFR1 and TNFR2 have also been observed in other lymphocyte subsets. As an example, in CD4+ T cells, these two receptors oppose each other in the maintenance of autoimmunity. TNFR1 promotes the differentiation of inflammatory subsets of T cells, including Th1 and Th17, to exacerbate autoimmunity, whereas TNFR2 promotes the differentiation and function of protective Tregs⁴⁴. Curiously, CD8+ T cells appear different to our results in NK cells, whereby TNFR2 induces cell death through overactivation, and TNFR1 limits this response to promote survival⁴⁵. In contrast, in myeloid cells such as macrophages TNF acting through TNFR1 is a well characterized trigger of apoptosis and necroptosis⁴⁶. However, TNFR2 can also sensitize macrophages towards being receptive to necroptosis in response to TNFR1 signaling⁴⁷. In neutrophils, both TNFR’s promote neutrophil driven clearance of bacterial skin infection, where TNFR1 is responsible for trafficking of neutrophils to the site of infection but TNFR2 is responsible for direct antibacterial function⁴⁸. Taken together, clearly TNFR1 and TNFR2 play different roles in different immune cells, and more work is required to tease apart the relative contribution of each receptor in various cell types.”*

Response to reviewer comments (Second round)

Manuscript No.: NCOMMS-23-28763A

Title: Dichotomous outcomes of TNFR1 and 2 signaling in NK cell-mediated immune responses during inflammation

Article Type: Article

Reviewer #1

The authors have adequately addressed this reviewer's comments.

Reviewer #2

The queries are answered and the experiments are performed by the Authors.

Reviewer #3

The authors have satisfactorily addressed most comments from the first round of review. I have only a few minor critiques/suggestions for this revised manuscript:

Comment 1

Please indicate in the Methods section that the CellCycleScoring function in Seurat was used for cell cycle analyses on the scRNAseq data.

Author response: The following text has been added to manuscript methods section:

"The CellCycleScoring function of Seurat was used for cell cycle analysis."

Comment 2

The text describing data from Fig. S4c-f (lines 126-129) should be revised to state that relative frequencies, not numbers, of Tim3-deficient and -overexpressing NK cells were analyzed in the adoptive transfer experiments. Since the data are depicted as relative frequencies, it is unclear whether Tim3-deficient NK cells are increasing or Tim3-overexpressing NK cells are decreasing in number (or both are occurring simultaneously), or both populations are increasing (or decreasing) in number but at different rates.

Author response: We agree that this is an important distinction. The text has been updated to the following:

"In adoptive transfer experiments, by transferring a 50:50 mix of Tim3-deficient and Tim3-overexpressing NK cells (Fig. 2c) we found that deletion of Tim3 lead to significant improvements in NK cell accumulation in the spleen and liver compared to Tim3 overexpressing cells, both in naïve mice and in the context of S. Typhimurium infection (Fig. 2d, Supplementary Fig. 5a,b)."

Comment 3

Please update legend for the blue histogram in Fig. S6g to reflect the “IL15+TNF (10ng/mL)” condition.

Author response: This change has been made in the figure to more accurately depict this experimental group.

Comment 4

For data in Figure 4b, please describe how “Survival” was calculated in the Methods section.

Author response: The following description has been added to the methods section:

“NK cell survival and proliferation were determined by calculating the cohort number and mean division number, respectively, as previously described^{28,67}. CTV peaks were given a generation number, i , where undivided cells were given $i = 0$ and increasing by 1 for each subsequent peak. Total cohort number was then determined to estimate NK cell death by accounting for the confounding influence of proliferation. To determine the total cohort number the number of each of cells within generation i was divided by 2^i to generate the cohort number for generation i . The total cohort number was then calculated by summing the cohort numbers for each generation:

$$\text{Total cohort number} = \sum \frac{\text{cell number}_i}{2^i}$$

Each biological replicate was then normalized to its corresponding IL-15 only control to obtain a relative survival value.

NK cell mean division number was determined by calculating the average number of divisions in each condition. Each generation number i is multiplied by the fraction of the total cohort which has undergone i divisions. The value of each generation was then summed to generate the mean division number:

$$\text{Mean division number} = \sum \left(i \times \frac{\text{cohort number}_i}{\text{total cohort number}} \right).$$

Comment 5

For data in Fig. S8, the legend states that NK cells were “analyzed three days post onset of visible symptoms”. Please clearly describe the specific visible symptoms that were used to define disease onset and clarify whether cells were collected from all mice 3 days after any single mouse displayed disease onset, or whether collection was relative to disease onset for each individual mouse.

Author response: *The figure legend has been updated to the following:*

*“Wild-type C57BL/6 mice were infected with wild-type *S. Typhimurium* (SL1344) by oral gavage and NK cells analyzed from all animals three days post onset of weight loss was observed in any mouse”*

Comment 6

For all text following line 231, “Tnfr1fl/fl” and “Tnfr2fl/fl” mice should be referred to as NcrcreTnfr1fl/fl and NcrcreTnfr2fl/fl, respectively (or Tnfr1-deficient and Tnfr2-deficient NK cells).

Author response: All text and figures have been updated accordingly.

Comment 7

All text that states or implies that TNFR1 and/or TNFR2 act individually to promote NK cell proliferation should be carefully revised to remove this interpretation, since the functional assays in Fig. S15 (Edu incorporation and Ki67 protein expression) do not support this conclusion. Examples include statements in lines 289, 292, and 304.

Author response: We agree that this interpretation is not supported by the functional assays, and thus all text has been revised to remove this interpretation.